# BIRDSET: A LARGE-SCALE DATASET FOR AUDIO CLASSIFICATION IN AVIAN BIOACOUSTICS

**Lukas Rauch**[1*] **Raphael Schwinger**[2] **Moritz Wirth**[1,3] **René Heinrich**[1,3] **Denis Huseljic**[1]
**Marek Herde**[1] **Jonas Lange**[2] **Stefan Kahl**[4] **Bernhard Sick**[1] **Sven Tomforde**[2] **Christoph Scholz**[1,3]
[1]University of Kassel [2]Kiel University [3]Fraunhofer IEE [4]TU Chemnitz    *lukas.rauch@uni-kassel.de

## ABSTRACT

Deep learning (DL) has greatly advanced audio classification, yet the field is limited by the scarcity of large-scale benchmark datasets that have propelled progress in other domains. While AudioSet is a pivotal step to bridge this gap as a universal-domain dataset, its restricted accessibility and limited range of evaluation use cases challenge its role as the sole resource. Therefore, we introduce BirdSet, a large-scale benchmark dataset for audio classification focusing on avian bioacoustics. BirdSet surpasses AudioSet with over 6,800 recording hours ($\uparrow 17\%$) from nearly 10,000 classes ($\uparrow 18\times$) for training and more than 400 hours ($\uparrow 7\times$) across eight strongly labeled evaluation datasets. It serves as a versatile resource for use cases such as multi-label classification, covariate shift, or self-supervised learning. We benchmark six well-known DL models in multi-label classification across three distinct training scenarios and outline further evaluation use cases in audio classification. We host our dataset on Hugging Face for easy accessibility and offer an extensive codebase to reproduce our results.

## 1 INTRODUCTION

Audio classification is critical in many domains such as environmental (Piczak, 2015) and wildlife monitoring (Kahl et al., 2021b). Audio data presents unique challenges for deep learning (DL), including low signal-to-noise ratios, temporal dependencies of events, and recording variability (Purwins et al., 2019). These challenges demand robust models capable of handling diverse evaluation use cases in multi-label classification (Fonseca et al., 2021), covariate shifts (changes in environments or recording devices) (Abeßer, 2020), class imbalance (few-shot learning) (Heggan et al., 2022), or label noise (annotation errors or weak labels) (Iqbal et al., 2022). However, large-scale datasets remain limited in audio classification compared to speech recognition or computer vision. While AudioSet (Gemmeke et al., 2017) offers substantial training data with over 5,800 recording hours, its restricted accessibility that requires manual data retrieval, lack of diverse evaluation scenarios and test datasets (Wang et al., 2021), and concerns regarding transferability to real-world environmental domains (Ghani et al., 2023) challenge its role as the only training resource. Thus, advancing audio classification requires not only universal datasets but also domain-specific data that offer a range of evaluation use cases for benchmarking the robustness and generalization performance of DL models.

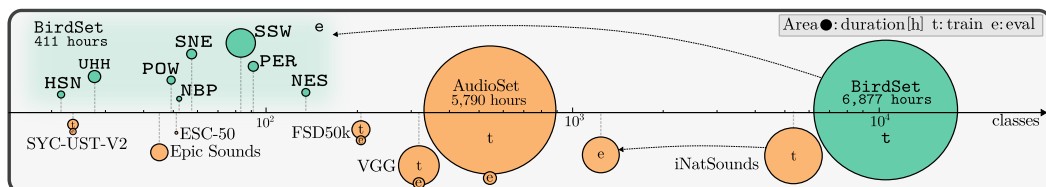

Figure 1: BirdSet's volume compared to broader audio classification datasets. The area of the circles represents the total recording duration in [h]. More details can be found in Appendix B.

Avian bioacoustics is well-suited as such a domain-specific application in audio classification due to (1) cost-effective data collection through passive acoustic monitoring (PAM) (Ross et al., 2023), (2) community-driven platforms like Xeno-Canto (XC) (Vellinga & Planqué, 2015) with an abundance of annotated recordings, and (3) the high complexity of bird vocalizations. This complexity reflects

diverse challenges relevant to broader audio classification, including diverse acoustic environments, extensive class diversity, variations in recording devices, and notable sound overlaps. The primary task in avian bioacoustics is the multi-label classification in PAM, comparable to classification in AudioSet. This serves as a crucial application since fluctuations in bird populations indicate broader shifts in biodiversity (Sekercioglu et al., 2016). Despite growing interest in computational avian bioacoustics (Stowell, 2021), there is no large-scale, easily accessible, and curated dataset available, hindering comparability across studies and creating barriers to accessibility from the broader audio domain (Rauch et al., 2023b). To advance audio classification and address the lack of a standardized benchmark in avian bioacoustics, we introduce the `BirdSet` benchmark dataset - a large-scale collection of bird vocalizations and a versatile resource for broader audio classification. `BirdSet` surpasses AudioSet in dataset volume and offers a unique test dataset collection featuring recordings from diverse regions (cf. Figure 1). We outline `BirdSet` and its contributions in the following:

---

**`BirdSet`: Outline and Contributions**

(1) We introduce the `BirdSet` **dataset collection**[a] on Hugging Face (HF) (Lhoest et al., 2021), featuring about 520,000 unique global bird sound recordings from nearly 10,000 species with over 6,800 hours for training and over 400 hours of PAM recordings with 170,000 annotated vocalizations across eight unique project sites for evaluation.

(2) `BirdSet` serves as an extensive **multi-purpose dataset** in audio classification with evaluation **use cases** such as self-supervised learning, event detection, multi-label classification under covariate and domain shifts with noisy labels or few-shot, and active learning.

(3) By providing a **large-scale train dataset** and a **diverse test dataset collection**, `BirdSet` represents a comprehensive and additional resource in audio classification (cf. Figure 1).

(4) A comprehensive **literature analysis** identifies and discusses **challenges** in computational avian bioacoustics embedded within the broader audio classification domain. We structure them to offer research guidelines and evaluation use cases resulting from `BirdSet`.

(5) We **benchmark** multi-label classification under covariate shift with noisy labels and task shift using well-known DL models. Our extensive empirical study evaluates distinct supervised training scenarios, including large-scale training and fine-tuning on `BirdSet`.

(6) An extensive **codebase**[b] with standardized training and evaluation protocols enables reproducing our results, supporting `BirdSet`'s utility, and easing accessibility for newcomers.

---

[a]`https://huggingface.co/datasets/DBD-research-group/BirdSet`
[b]`https://github.com/DBD-research-group/BirdSet`

## 2 Current Challenges and Related Work

Avian bioacoustics exemplifies challenges in audio classification, including managing diverse and noisy acoustic environments or dealing with class imbalance. Thus, it is our primary case study for illustrating real-world evaluation use cases in the field represented in `BirdSet`'s datasets. In this section, we outline these challenges, review how related work addresses them, and outline our approach to tackling them in our benchmark, highlighting how `BirdSet` differs from related datasets. Detailed explanations of our approaches are provided in Section 3 and Section 4.

### 2.1 Challenge 1: Datasets

**Challenge description.** Audio data exhibits complex characteristics, including variable lengths, overlapping signals, and variability in recording sources (e.g., recording type or device). As a domain-specific audio classification task, avian bioacoustics exemplifies and adds to these complexities, making it suitable for exploring real-world audio challenges. Avian bioacoustics differentiates between focal and soundscape recordings (Kahl et al., 2021b). *Focal recordings* involve a recordist aiming a directional microphone toward the source of bird vocalizations (i.e., sound events), capturing sequences of calls from primary and occasionally secondary species, which typically results in a multi-class problem. Their abundance and variability on citizen-science platforms such as XC make them particularly suitable as training data. However, they do not represent entire acoustic environments (i.e., soundscapes) and are usually weakly labeled without specific vocalization times, making them

unsuitable for evaluation in PAM (Van Merriënboer et al., 2024). *Soundscape recordings* in PAM are passively collected by omnidirectional microphones within a static area over extended periods (Kahl et al., 2021b), capturing bird vocalizations alongside environmental noise with minimal habitat disruption. Often strongly labeled from BirdCLEF competitions (Kahl et al., 2022b), soundscapes offer a comprehensive audio representation of a real-world domain, making them ideal for testing. Due to the simultaneous occurrence of multiple sounds, soundscapes reflect a multi-label problem. However, their static nature, limited geographical coverage, and high labeling cost render them unsuitable for large-scale model training (Van Merriënboer et al., 2024).

Table 1: Datasets employed in current bird sound classification publications. Model training is analyzed by the task (multi-label ●●● or multi-class ●●), architecture, and input type.

| Sources | | Focals | | | | | Soundscapes | | | | | | | | | | Task | | Model | | Input | |
|---|---|---|---|---|---|---|---|---|---|---|---|---|---|---|---|---|---|---|---|---|---|---|
| | | XC | INA | MAC | CBI | BD | HSN | SNE | UHH | PER | SSW | POW | CAP | NBP | S2L | VOX | ●●● | ●● | CNN | Trnsf | Spec | Wave |
| Bellafkir et al. | train | ✓ | ✗ | ✗ | ✗ | ✗ | ✗ | ✗ | ✗ | ✗ | ✗ | ✗ | ✗ | ✗ | ✗ | ✓ | ✓ | ✗ | ✓ | ✗ | ✓ | ✗ |
| | eval | ✓ | ✗ | ✗ | ✗ | ✗ | ✗ | ✗ | ✗ | ✗ | ✗ | ✗ | ✗ | ✗ | ✗ | ✗ | | | | | | |
| Bellafkir et al. | train | ✓ | ✓ | ✗ | ✗ | ✗ | ✗ | ✗ | ✗ | ✗ | ✗ | ✗ | ✗ | ✗ | ✗ | ✗ | ✓ | ✗ | ✗ | ✓ | ✓ | ✗ |
| | eval | ✗ | ✗ | ✗ | ✗ | ✗ | ✗ | ✗ | ✗ | ✗ | ✗ | ✗ | ✗ | ✗ | ✗ | ✗ | | | | | | |
| Bravo Sanchez et al. | train | ✗ | ✗ | ✗ | ✗ | ✗ | ✗ | ✗ | ✗ | ✗ | ✗ | ✗ | ✗ | ✓ | ✗ | ✗ | ✗ | ✓ | ✓ | ✗ | ✗ | ✓ |
| | eval | ✗ | ✗ | ✗ | ✓ | ✗ | ✗ | ✗ | ✗ | ✗ | ✗ | ✗ | ✗ | ✗ | ✗ | ✗ | | | | | | |
| Clark et al. | train | ✓ | ✗ | ✗ | ✓ | ✗ | ✗ | ✗ | ✗ | ✗ | ✗ | ✗ | ✗ | ✗ | ✗ | ✗ | ✓ | ✗ | ✓ | ✗ | ✓ | ✗ |
| | eval | ✗ | ✗ | ✗ | ✗ | ✗ | ✗ | ✗ | ✗ | ✗ | ✗ | ✗ | ✗ | ✓ | ✗ | ✗ | | | | | | |
| Denton et al. | train | ✓ | ✗ | ✓ | ✗ | ✗ | ✗ | ✗ | ✗ | ✗ | ✗ | ✗ | ✗ | ✗ | ✗ | ✓ | ✓ | ✗ | ✓ | ✗ | ✓ | ✗ |
| | eval | ✗ | ✗ | ✗ | ✗ | ✗ | ✓ | ✗ | ✗ | ✓ | ✗ | ✓ | ✗ | ✗ | ✗ | ✗ | | | | | | |
| Eichinski et al. | train | ✗ | ✗ | ✗ | ✗ | ✗ | ✗ | ✗ | ✗ | ✗ | ✗ | ✗ | ✗ | ✗ | ✗ | ✗ | ✗ | ✓ | ✓ | ✗ | ✓ | ✗ |
| | eval | ✗ | ✗ | ✗ | ✗ | ✗ | ✗ | ✗ | ✗ | ✗ | ✗ | ✗ | ✗ | ✗ | ✗ | ✗ | | | | | | |
| Fu et al. | train | ✓ | ✗ | ✗ | ✗ | ✗ | ✗ | ✗ | ✗ | ✗ | ✗ | ✗ | ✗ | ✗ | ✗ | ✗ | ✓ | ✗ | ✓ | ✗ | ✓ | ✗ |
| | eval | ✓ | ✗ | ✗ | ✗ | ✗ | ✗ | ✗ | ✗ | ✗ | ✗ | ✗ | ✗ | ✗ | ✗ | ✗ | | | | | | |
| Gupta et al. | train | ✗ | ✗ | ✗ | ✓ | ✗ | ✗ | ✗ | ✗ | ✗ | ✗ | ✗ | ✗ | ✗ | ✗ | ✗ | ✗ | ✓ | ✓ | ✗ | ✓ | ✗ |
| | eval | ✗ | ✗ | ✗ | ✓ | ✗ | ✗ | ✗ | ✗ | ✗ | ✗ | ✗ | ✗ | ✗ | ✗ | ✗ | | | | | | |
| Hamer et al. | train | ✓ | ✗ | ✗ | ✗ | ✗ | ✗ | ✗ | ✗ | ✗ | ✗ | ✗ | ✗ | ✗ | ✗ | ✓ | ✓ | ✗ | ✓ | ✗ | ✓ | ✗ |
| | eval | ✗ | ✗ | ✗ | ✗ | ✗ | ✓ | ✓ | ✓ | ✓ | ✓ | ✓ | ✓ | ✗ | ✗ | ✗ | | | | | | |
| Höchst et al. | train | ✓ | ✓ | ✗ | ✗ | ✗ | ✗ | ✗ | ✗ | ✗ | ✗ | ✗ | ✗ | ✗ | ✗ | ✗ | ✓ | ✗ | ✓ | ✗ | ✓ | ✗ |
| | eval | ✓ | ✓ | ✗ | ✗ | ✗ | ✗ | ✗ | ✗ | ✗ | ✗ | ✗ | ✗ | ✗ | ✗ | ✗ | | | | | | |
| Hu et al. | train | ✓ | ✗ | ✗ | ✗ | ✓ | ✗ | ✗ | ✗ | ✗ | ✗ | ✗ | ✗ | ✗ | ✗ | ✗ | ✗ | ✓ | ✓ | ✗ | ✓ | ✗ |
| | eval | ✓ | ✗ | ✗ | ✗ | ✓ | ✗ | ✗ | ✗ | ✗ | ✗ | ✗ | ✗ | ✗ | ✗ | ✗ | | | | | | |
| Jeantet & Dufourq | train | ✓ | ✗ | ✗ | ✗ | ✗ | ✗ | ✗ | ✗ | ✗ | ✗ | ✗ | ✗ | ✗ | ✗ | ✗ | ✗ | ✓ | ✓ | ✗ | ✓ | ✗ |
| | eval | ✓ | ✗ | ✗ | ✗ | ✗ | ✗ | ✗ | ✗ | ✗ | ✗ | ✗ | ✗ | ✗ | ✗ | ✗ | | | | | | |
| Kahl et al. | train | ✓ | ✗ | ✓ | ✗ | ✗ | ✗ | ✗ | ✗ | ✗ | ✗ | ✗ | ✗ | ✗ | ✗ | ✓ | ✓ | ✗ | ✓ | ✗ | ✓ | ✗ |
| | eval | ✓ | ✗ | ✗ | ✗ | ✗ | ✗ | ✗ | ✗ | ✓ | ✗ | ✗ | ✗ | ✗ | ✗ | ✗ | | | | | | |
| Liu et al. | train | ✓ | ✗ | ✗ | ✗ | ✗ | ✗ | ✗ | ✗ | ✗ | ✗ | ✗ | ✗ | ✗ | ✗ | ✗ | ✗ | ✓ | ✓ | ✗ | ✓ | ✗ |
| | eval | ✓ | ✗ | ✗ | ✗ | ✗ | ✗ | ✗ | ✗ | ✗ | ✗ | ✗ | ✗ | ✗ | ✗ | ✗ | | | | | | |
| Liu et al. | train | ✓ | ✗ | ✗ | ✗ | ✗ | ✗ | ✗ | ✗ | ✗ | ✗ | ✗ | ✗ | ✗ | ✗ | ✗ | ✗ | ✓ | ✓ | ✗ | ✓ | ✗ |
| | eval | ✓ | ✗ | ✗ | ✗ | ✗ | ✗ | ✗ | ✗ | ✗ | ✗ | ✗ | ✗ | ✗ | ✗ | ✗ | | | | | | |
| Swaminathan et al. | train | ✓ | ✗ | ✗ | ✗ | ✗ | ✗ | ✗ | ✗ | ✗ | ✗ | ✗ | ✗ | ✗ | ✗ | ✗ | ✓ | ✗ | ✗ | ✓ | ✗ | ✓ |
| | eval | ✓ | ✗ | ✗ | ✗ | ✗ | ✗ | ✗ | ✗ | ✗ | ✗ | ✗ | ✗ | ✗ | ✗ | ✗ | | | | | | |
| Tang et al. | train | ✓ | ✗ | ✗ | ✗ | ✓ | ✗ | ✗ | ✗ | ✗ | ✗ | ✗ | ✗ | ✗ | ✗ | ✗ | ✗ | ✓ | ✗ | ✓ | ✓ | ✗ |
| | eval | ✓ | ✗ | ✗ | ✗ | ✗ | ✗ | ✗ | ✗ | ✗ | ✗ | ✗ | ✗ | ✗ | ✗ | ✗ | | | | | | |
| Wang et al. | train | ✓ | ✗ | ✗ | ✗ | ✗ | ✗ | ✗ | ✗ | ✗ | ✗ | ✗ | ✗ | ✗ | ✗ | ✗ | ✗ | ✓ | ✗ | ✗ | ✓ | ✗ |
| | eval | ✓ | ✗ | ✗ | ✗ | ✗ | ✗ | ✗ | ✗ | ✗ | ✗ | ✗ | ✗ | ✗ | ✗ | ✗ | | | | | | |
| Xiao et al. | train | ✗ | ✗ | ✗ | ✗ | ✓ | ✗ | ✗ | ✗ | ✗ | ✗ | ✗ | ✗ | ✗ | ✗ | ✗ | ✗ | ✓ | ✓ | ✗ | ✓ | ✗ |
| | eval | ✗ | ✗ | ✗ | ✗ | ✗ | ✗ | ✗ | ✗ | ✗ | ✗ | ✗ | ✗ | ✗ | ✗ | ✗ | | | | | | |
| Xie & Zhu | train | ✓ | ✗ | ✗ | ✗ | ✗ | ✗ | ✗ | ✗ | ✗ | ✗ | ✗ | ✗ | ✗ | ✗ | ✗ | ✗ | ✓ | ✓ | ✗ | ✓ | ✗ |
| | eval | ✓ | ✗ | ✗ | ✗ | ✗ | ✗ | ✗ | ✗ | ✗ | ✗ | ✗ | ✗ | ✗ | ✗ | ✗ | | | | | | |
| Zhang et al. | train | ✗ | ✗ | ✗ | ✗ | ✓ | ✗ | ✗ | ✗ | ✗ | ✗ | ✗ | ✗ | ✗ | ✗ | ✗ | ✗ | ✓ | ✓ | ✗ | ✓ | ✗ |
| | eval | ✗ | ✗ | ✗ | ✗ | ✗ | ✗ | ✗ | ✗ | ✗ | ✗ | ✗ | ✗ | ✗ | ✗ | ✗ | | | | | | |
| BirdSet | train | ✓ | ✗ | ✗ | ✗ | ✗ | ✗ | ✗ | ✗ | ✗ | ✗ | ✗ | ✗ | ✗ | ✗ | ✓ | ✓ | ✗ | ✓ | ✓ | ✓ | ✓ |
| | eval | ✗ | ✗ | ✗ | ✗ | ✗ | ✓ | ✓ | ✓ | ✓ | ✓ | ✓ | ✓ | ✗ | ✓ | ✗ | | | | | | |

**Related work.** Studies in audio classification commonly employ AudioSet for large-scale training and smaller datasets such as ESC-50 (Piczak, 2015) for fine-tuning and testing (Gong et al., 2021; Huang et al., 2022; Chen et al., 2023). However, these datasets cannot represent all real-world complexities (e.g., class imbalance) due to limited size and quantity compared to vision datasets (e.g., ImageNet (Deng et al., 2009)). Despite the wealth of data in avian bioacoustics (Vellinga & Planqué, 2015), the absence of standardized datasets similar to AudioSet results in inconsistent data selections and processing, outlined in Table 1. This lack of standardization affects comparability and complicates accessibility to this field. Studies usually create custom train and test datasets of focal recordings from XC, failing to generalize to realistic PAM scenarios with soundscapes. Additionally, using weak labels for testing compromises the practical result validity (Fonseca et al., 2021), and the dynamic nature of XC further complicates reproducibility and comparability across studies. Other research directions operate in PAM scenarios by evaluating an arbitrary set of soundscapes (Denton et al., 2022; Bellafkir et al., 2023) with custom training datasets from XC that are not publicly available. In contrast, BirdSet eliminates dataset processing and collection by providing the first curated collection with uniform metadata (e.g., label formatting) via HF. It includes an extensive volume of focals for training and a collection of soundscapes for testing, enabling model performance evaluation across various regions in PAM and facilitating comparability across studies.

**Related benchmarks.** AudioSet is a universal-domain dataset for large-scale audio classification, featuring approximately 2.1 million samples across 527 classes, totaling 5,800 hours. While it is the only environmental audio classification dataset comparable in scope to BirdSet, it only includes one small test subset and requires manual retrieval of audio files. In contrast, BirdSet includes nearly 10,000 domain-specific classes, over 6,800 hours of training data, and eight distinct and

strongly labeled test datasets. Its easy usability through HF contrasts with the complex setup required for accessing AudioSet, further highlighting `BirdSet`'s benefits. ESC-50, a smaller dataset with 2,000 recordings, primarily serves for small-scale evaluation or fine-tuning. FSDK50 (Fonseca et al., 2021) offers 50,000 curated clips across 200 general-domain classes from the AudioSet ontology to improve accessibility, though it does not match `BirdSet`'s volume. In avian bioacoustics, the BirdCLEF challenge (Kahl et al., 2023) regularly introduces novel, strongly labeled soundscape datasets for testing. However, its competition format emphasizes score optimization over broader research, and the varying datasets and metrics across editions limit its suitability for benchmarking. In contrast, `BirdSet` offers a curated collection with standardized training and evaluation protocols, enabling comparisons across studies and complementing BirdCLEF as a foundation for future challenges. The BEANS (Hagiwara et al., 2023) and BIRB (Hamer et al., 2023) benchmarks mark a shift towards structured datasets in bioacoustics but face issues in dataset diversity and accessibility. While BEANS covers various animal sounds, it lacks volume and does not provide a large-scale training dataset suitable for representation learning like `BirdSet`, limiting its evaluation use cases. Its multi-class nature also falls short in assessing generalization for avian bioacoustics and broader environmental audio classification. In contrast, `BirdSet` offers a larger, more diverse, and versatile benchmark, with circa 10,000 bird species globally for training and eight distinct evaluation locations. Its segment-based multi-label classification of soundscapes better mirrors real-world PAM and audio classification conditions. While BIRB and `BirdSet` focus on bird data, their task and scope differ significantly. `BirdSet` offers a readily accessible multi-label classification task, bridging broader audio classification with various evaluation use cases. In contrast, BIRB provides a retrieval task centered on avian vocalizations, where pre-trained bioacoustic models are evaluated by ranking species-specific vocalizations based on a few examples. Additionally, BIRB requires manual data processing and user-defined query inputs to create datasets, hindering cross-study comparisons and limiting its applicability beyond bioacoustics. `BirdSet`'s simplicity and versatility make it a more user-friendly benchmark for avian bioacoustics and general audio classification.

## 2.2 CHALLENGE 2: MODEL TRAINING

**Challenge description.** Audio classification involves detecting and handling overlapping events in a multi-label context, typically associated with weak file-level labels, to reduce annotation effort (Fonseca et al., 2021). This complexity is often reduced to a more straightforward multi-class task in datasets such as ESC-50 (Piczak, 2015), potentially limiting a model's ability to handle overlapping sounds and event ambiguity. In computational avian bioacoustics, tasks range from detection to identification of bird species, primarily focusing on species classification through vocalizations (Van Merriënboer et al., 2024), nearly identical to general audio classification. Research typically employs multi-class classification for focal recordings and multi-label classification for soundscapes. Recent studies follow a typical training process (Stowell, 2021), as outlined in Figure 2. It starts by detecting bird vocalizations in weakly labeled focals, enabling a recording to yield multiple samples. Comparable to audio classification, vocalizations are then converted into spectrogram images which visualize frequency intensity, reducing noise and preparing the data for classification with vision-based DL architectures (Kahl et al., 2021b). This conversion requires manual parameter tuning (e.g., frequency resolution), complicating model comparability (Gazneli et al., 2022; Rauch et al., 2023b). Augmentation techniques applied to waveforms or spectrograms enhance data variety and align the training data distribution of focal recordings more closely to the test data distribution (i.e., soundscape recordings in PAM), aiding in model generalization.

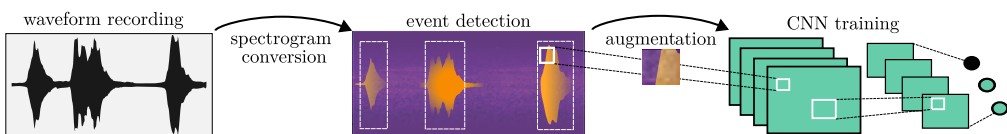

Figure 2: Typical model training for classification in avian bioacoustics.

**Related work.** Most audio classification and avian bioacoustics research leverages spectrogram images as inputs, as shown in Table 1. While general audio classification increasingly adopts transformer architectures (Chen et al., 2023), CNNs still dominate avian bioacoustics (Kahl et al., 2021b). Bellafkir et al. (2024) use a variant of the vision-based audio spectrogram transformer

(AST) (Gong et al., 2021). Bravo Sanchez et al. (2021) and Swaminathan et al. (2024) demonstrate the potential of learning directly from raw waveforms by employing SincNet (Ravanelli & Bengio, 2019) and Wav2Vec 2.0 (W2V2) (Baevski et al., 2020). While audio classification has benefited from self-supervised learning (Chen et al., 2023; 2024)), avian bioacoustics has only begun initial explorations (Bellafkir et al., 2024; Moummad et al., 2024). Current state-of-the-art (SOTA) models BirdNET (Kahl et al., 2021b) and Google's Perch (Hamer et al., 2023) still rely on conventional supervised learning with EfficientNet (Tan & Le, 2020). However, these large audio classification models, trained on approximately 10,000 bird species from XC, produce valuable embeddings for downstream tasks in avian bioacoustics. In contrast, models pre-trained on general domain audio exhibit poor transferability to the domain (Ghani et al., 2023; Nolan et al., 2023). Additionally, multi-label classification remains underrepresented in current work despite its importance in practical PAM scenarios. Our benchmark addresses this gap by introducing three training protocols for multi-label audio classification featuring large-scale representation learning and fine-tuning. We are the first to present benchmark results for the current SOTA in avian bioacoustics, extending beyond typical convolutional neural network (CNN) architectures to include transformers (e.g., AST) or SOTA CNNs (ConvNext (Liu et al., 2022c)). Our supervised training protocols are designed to support future evaluation use cases in self-supervised representation learning, including large-scale pre-training and fine-tuning on `BirdSet`. All models and benchmarks are open-sourced via HF.

## 2.3 CHALLENGE 3: MODEL ROBUSTNESS

**Challenge description.** A robust audio classification model must generalize across diverse acoustic environments, varying notably by their location. It requires handling recording variations, including noise, sound source distance, or recording type. Avian bioacoustics under PAM conditions exemplifies this with complex and diverse bird vocalizations (Byers & Kroodsma, 2016). A bird species emits multiple distinct vocalizations, including songs, call types, or regional dialects, blending into the complex dawn chorus (Berg et al., 2006). We identified four key challenges in `BirdSet` that hinder achieving robustness in real-world audio classification tasks based on avian bioacoustics, illustrated in Figure 3. We analyze how related work and our multi-label benchmark address them in the following. We also outline how `BirdSet` offers opportunities to evaluate them for audio classification.

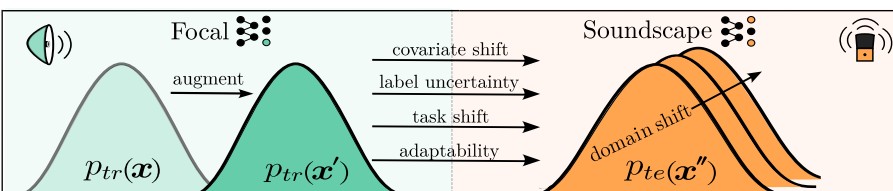

Figure 3: Key obstacles for model robustness in bird sound classification.

**Related work - covariate and domain shift.** Covariate and domain shifts occur when training and test input distributions differ (Sugiyama & Kawanabe, 2012), often due to varying recording conditions and devices. Evaluating these shifts in audio classification is challenging because most datasets are highly processed and lack variability in the test data (Fonseca et al., 2021). This gap is particularly pronounced in avian bioacoustics due to the use of focals for training and soundscapes for testing. Moreover, the sound profiles of recordings vary due to differences in recording devices, environmental conditions, or proximity to the sound source. As shown in Table 1, current research typically creates custom train and test datasets from XC, isolating vocalization events in advance. However, this approach bypasses the shift encountered in real-world conditions, where focal-trained models must generalize to soundscape recordings (Kahl et al., 2021b). In a PAM setting, research employs augmentations by adding background sounds from external soundscapes (Hamer et al., 2023; Denton et al., 2022) (e.g., from VOX (Stowell et al., 2019)) or noise to align the train and test distributions more closely. Supervised pre-training on various species outside the test label distribution has enhanced model robustness under domain and covariate shift (Clark et al., 2023). Studies also incorporate soundscapes from a specific use case into training to address covariate shift (Eichinski et al., 2022; Höchst et al., 2022), challenging adaption to new PAM locations. `BirdSet` provides different training scenarios for focal-based training datasets that incorporate a supervised pre-training procedure and a set of augmentations to align the train and test distributions more closely.

**Related work - label uncertainty.** Label uncertainty or noisy labels is a common challenge in audio classification, occurring from weakly labeled recordings that provide only file-level labels without precise event timestamps or from labeling errors in large datasets (Fonseca et al., 2021). This is particularly problematic for evaluation, as weak test labels can significantly impact benchmarking accuracy (e.g., in AudioSet (Fonseca et al., 2021)). In avian bioacoustics, focal recordings from XC are also usually weakly labeled, lacking exact timestamps for the bird vocalization label. However, the citizen-science aspect of XC helps reduce labeling errors. In contrast, various publicly available soundscape datasets include precise annotations by ornithologists (Hopping et al., 2022). Ambiguity also arises due to the lack of specific labels for different vocalization types, as different vocalizations of a species are assigned to the same label. Therefore, a reliable audio model must handle noisy labels in PAM. Current work typically extracts potential events in focal recordings through pre-processing. The most straightforward approach for event detection assumes bird vocalizations occur at the start of recordings (Gupta et al., 2021), ignoring temporal variations. Alternatively, a peak detection algorithm identifies vocalization events in a recording (Denton et al., 2022; Hamer et al., 2023). However, these labels remain ambiguous, as an event could originate from another sound source. Other methods include random selection from a recording (Bellafkir et al., 2024; Eichinski et al., 2022) or dedicated detection models (Clark et al., 2023; Bellafkir et al., 2023). Additionally, curated clips of recordings are employed where label noise is removed beforehand (Tang et al., 2023; Hu et al., 2023; Jeantet & Dufourq, 2023), comparable to audio classification datasets such as AudioSet. Advanced techniques include combining clustering with peak detection (Michaud et al., 2023), tracking feature changes through self-supervised learning (Bermant et al., 2022), or isolating vocalizations via unsupervised sound separation (Denton et al., 2022). `BirdSet` provides a comprehensive training dataset from XC with minimal label errors, featuring detected vocalization events and clusters to address label uncertainty but keeping event usage flexible for further research. Moreover, the test dataset collection exclusively contains strong labels, ensuring high-quality evaluation.

**Related work - task shift.** In audio classification, pre-training on multi-label datasets like AudioSet and fine-tuning on multi-class datasets like ESC-50 introduces a task shift from pre-training to testing (Huang et al., 2022). A similar shift occurs in avian bioacoustics, where focal-trained models are evaluated on soundscapes in PAM, moving the task from multi-class to multi-label classification. Most studies avoid the task shift by focusing on multi-class classification with manually filtered vocalizations in training and testing. Swaminathan et al. (2024) diverge from this trend by combining multiple XC focal recordings to mimic a multi-label scenario in training and testing. Denton et al. (2022), Bellafkir et al. (2023), and Kahl et al. (2021b) transition towards a more realistic segment-based evaluation of soundscape recordings. They address the task shift by augmenting the focal training data with label-mixup (Zhang et al., 2018), combining multiple recordings and labels into one instance, simulating a multi-label task in training. `BirdSet`'s training protocol uses focal recordings, while the evaluation protocol employs soundscapes, introducing an inherent task shift. To better align the training and test data distributions, we incorporate label-mixup.

**Related work - adaptability.** In real-world audio classification, models often operate in a few-shot setting with limited training data and a novel recording domain. This challenge is especially pronounced in avian bioacoustics due to many unique species and differing label distributions between focals and soundscapes (Van Merriënboer et al., 2024). While models trained on diverse audios have shown to generalize and transfer across various conditions (Ghani et al., 2023; Chen et al., 2023), they exhibit a subpopulation shift towards particular niches when fine-tuned for a specialized task in audio classification (Tran et al., 2022). Domain shifts between PAM environments in avian bioacoustics, such as variations in background sounds, further underscore the need for adaptable models. Currently, large-scale SOTA models (Hamer et al., 2023; Kahl et al., 2021b) rely on conventional supervised learning over extensive datasets that are utilized to obtain robust representations for few-shot learning and adaptation. Ghani et al. (2023) demonstrate that Perch excels in few-shot scenarios by fine-tuning only the classification layer. Methods to manage subpopulation shifts include limiting logits to relevant species in test datasets (Hamer et al., 2023) or training on tailored subsets to adapt to new domains (Denton et al., 2022). Bellafkir et al. (2024) and Rauch et al. (2024a) apply active learning to bird sound classification, addressing the limitations of static models in dynamic environments. `BirdSet` offers different training scenarios to evaluate model adaptability, including training on extensive datasets to create large-scale bird sound classification models and fine-tuning on tailored subsets for specific tasks.

## 2.4 CHALLENGE 4: EVALUATION

**Challenge description.** Evaluating models in audio classification is challenging due to the temporal ambiguity of audio events. Avian bioacoustics underscores this complexity, including various practical PAM scenarios and downstream tasks (e.g., event detection, density estimation, classification). In the detection and classification of bird vocalizations, we differentiate between event-based and segment-based evaluation (Van Merriënboer et al., 2024), as illustrated in Figure 4. *Event-based evaluation* isolates vocalizations (e.g., peak event detection) in the test dataset. Isolating and applying multi-class classification is similar to ESC-50 but cannot capture temporal dynamics in complex scenarios with overlapping sounds. *Segment-based evaluation* is the preferred method for assessing performance in realistic PAM settings. In this approach, long-duration soundscapes are segmented into fixed intervals, with labels assigned to each segment (Denton et al., 2022), comparable to AudioSet's classification task. This evaluation is challenged by ambiguous ground truth assignments in edge cases and validating results during training under covariate shift. Segments simplify the task to vocalization detection or multi-label classification, utilizing threshold-dependent or threshold-independent metrics.

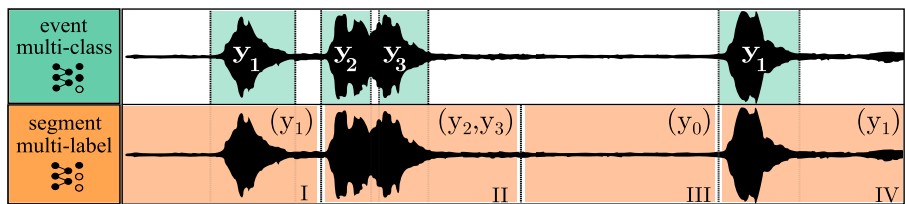

Figure 4: An illustration of segment- and event-based evaluations (Van Merriënboer et al., 2024).

**Related work.** When evaluating on focals, model performance is commonly assessed through isolated vocalization events, comparable to clips of ESC-50, and multi-class metrics such as precision or recall (Liu et al., 2022a; Jeantet & Dufourq, 2023). Soundscapes in PAM are typically evaluated via segment-based multi-label classification within five-second intervals (Denton et al., 2022; Hamer et al., 2023), similar to 10-second clips in AudioSet. Previous BirdCLEF competitions used the multi-label F1-Score (Kahl et al., 2021a; 2022b), which requires class-dependent threshold tuning, complicating comparability and hindering insights into model generalization. Following general multi-label audio classification (Gemmeke et al., 2017), recent work favors threshold-independent metrics. The class-based mean average precision (cmAP) calculates the average precision across thresholds per class, followed by macro averaging (Kahl et al., 2023). While it provides a macro view of the model's ability to rank positive over negative instances, it can be noisy for species with sparse labels (Denton et al., 2022). The area under the receiver operating characteristic curve (AUROC) measures a model's ability to rank a randomly selected positive over a negative instance (Van Merriënboer et al., 2024). Unlike cmAP, which is heavily influenced by the number of positives and negatives, AUROC is more robust to class imbalance across datasets, providing a baseline of 0.5 for random rankings (Hamer et al., 2023). Top-1 accuracy (T1-Acc) measures whether the highest predicted probability matches one of the correct classes in a segment (Denton et al., 2022), making it helpful in identifying a single species. Unlike ESC-50 or AudioSet, no standardized performance overview allows direct comparison across soundscapes. Thus, current research lacks a standardized evaluation protocol, complicating study comparisons. `BirdSet` introduces an evaluation protocol for multi-label audio classification, allowing researchers to evaluate and compare their models effectively. Our benchmark employs a collection of threshold-independent metrics.

## 3 BIRDSET: A LARGE-SCALE AUDIO DATASET COLLECTION

**Collection and curation.** `BirdSet` provides a large-scale collection of curated train and test datasets with varying complexities, targeting multi-label audio event classification of bird vocalizations. All focal training data originates from XC, the largest and most popular citizen-science platform for avian bioacoustics. We collected and curated legally compliant, high-quality recordings. The test datasets include a diverse set of PAM scenarios with soundscape recordings covering different difficulty levels, class diversity, and geographical variations. Ensuring all datasets are legally compliant and equipped with strong labels allows for high-quality evaluation and realistic simulation of train and test scenarios in PAM. Inspired by the works of (Wang et al., 2018; Rauch et al., 2023a), we aggregate

various datasets into one diverse test collection to achieve representative model generalization results. Our curation process to achieve a uniform metadata format includes unifying the label format of all train and test recordings using the eBird code taxonomy (Sullivan et al., 2009). Moreover, we processed soundscapes into slices for segment-based evaluation, aligned the label formats with short-range focals for multi-label classification, and provided vocalization events in focals through bambird (Michaud et al., 2023). Recordings are processed in a resolution of 32 kHz, capturing the frequency range of most bird vocalizations (Kahl et al., 2021b). We integrate the complete collection into one HF dataset with accompanying code to significantly simplify usability for researchers. More details are provided in Appendix B.

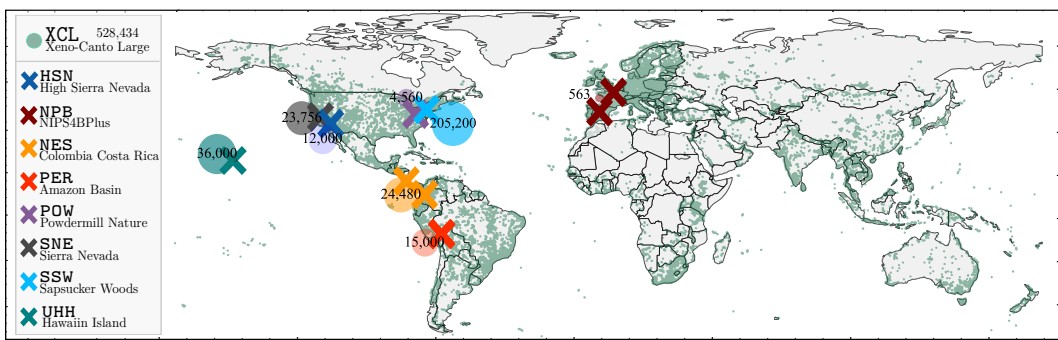

Figure 5: `BirdSet`'s geographical distribution of focals for training and soundscapes for testing.

**Datasets.** An overview of the `BirdSet` dataset collection is provided in Table 2 with a geographical distribution shown in Figure 5. We organize the datasets into three functional groups. *Train* comprises focal recordings suitable for large-scale training. Xeno-Canto large (XCL) covers a comprehensive snapshot of XC, featuring approximately 530,000 curated recordings across nearly 10,000 species. Since we equip the variable-length recordings with detected events, the training dataset size can be expanded based on the events extracted from each recording. XCL is `BirdSet`'s largest dataset that is comparable to the training datasets of current SOTA models BirdNET and Perch, but the first to be entirely publicly available. Xeno-Canto medium (XCM) is a specialized subset of XCL that includes only recordings from 409 unique species across our test datasets. `BirdSet` also provides non-bird soundscape recordings from the VOX dataset (Lostanlen

Table 2: Overview of datasets in `BirdSet`. $J$ denotes the evenness index.

| | Set | \|Train\| | \|Test\| | #Annot | $J$ | #$C$ |
|---|---|---|---|---|---|---|
| *Train* | XCL | 528,434 | ✗ | 528,343 | 0.85 | 9,734 |
| | XCM | 89,798 | ✗ | 89,798 | 0.94 | 409 |
| *Aux* | POW | 14,911 | 4,560 | 16,052 | 0.66 | 48 |
| | VOX | 20,331 | ✗ | ✗ | ✗ | ✗ |
| *Test & Ded. Fine-Tuning* | PER | 16,802 | 15,120 | 14,798 | 0.78 | 132 |
| | NES | 16,117 | 24,480 | 6,952 | 0.76 | 89 |
| | UHH | 3,626 | 36,637 | 59,583 | 0.64 | 25 |
| | HSN | 5,460 | 12,000 | 10,296 | 0.54 | 21 |
| | NBP | 24,327 | 563 | 5,493 | 0.92 | 51 |
| | SSW | 28,403 | 205,200 | 50,760 | 0.77 | 81 |
| | SNE | 19,390 | 23,756 | 20,147 | 0.70 | 56 |

et al., 2018b), which serve as background noise or no-call segments for augmentations. POW is a relatively small, fully annotated soundscape dataset to validate performance or tune hyperparameters for large-scale models trained on XCL or XCM. *Test & Dedicated Fine-Tuning* consists of fully annotated and high-quality soundscapes for evaluating multi-label classification approaches. Following related work (Denton et al., 2022), we divide each recording into 5-second segments and assign the labels based on ground truth timestamps. We attribute a label for vocalizations spanning multiple segments if the vocalization lasts over 0.5 seconds within a segment (Kahl et al., 2023). Each segment is an independent test sample containing none (0-vector), one or multiple species. Table 2 displays the number of segments and annotations, representing the overall recording duration and frequency of bird vocalizations. If the number of annotations exceeds the number of segments, it indicates overlapping vocalizations or concentrated activity in a few segments, with others remaining empty. We also offer dedicated training subsets for each test dataset that include only recordings of species from XCL present in the respective set. They serve as fine-grained training or fine-tuning datasets. We present the test datasets' class imbalance through Pielou's evenness index $J$ (Pielou, 1966) by calculating the relative class entropy per dataset. A value of 1 signifies perfect balance, while 0 indicates a significant imbalance across species.

## 4 BENCHMARK: MULTI-LABEL CLASSIFICATION

This section presents our multi-label audio classification benchmark as the primary use case for `BirdSet`. In this benchmark, robust models must handle `BirdSet`'s inherent challenges that include covariate shift (train and test distributions), label uncertainty (weak labels), task shift (multi-class and multi-label), class imbalance, and subpopulation shift (adapting to a species subset). Detailed experimental settings and results are provided in Appendix C.

### 4.1 EXPERIMENTAL SETUP

**Training protocol.** We explore three supervised multi-label training scenarios relevant to current audio classification research (Hamer et al., 2023; Ghani et al., 2023; Purwins et al., 2019). We one-hot encode the focal recordings during training to align their labels with the multi-label soundscapes used in testing. In **large training** (**LT**) and **medium training** (**MT**), we train models on XCL and XCM, respectively. This results in supervised representation learning for various bird species in diverse environments, comparable to a pre-trained model on AudioSet. In **dedicated fine-tuning** (**DT**), we train the models on the specific subsets, adapting them to specialized tasks, similar to fine-tuning a pre-trained AudioSet model on ESC-50. We employ five well-known audio model architectures. This includes spectrogram-based models: EfficientNet (Tan & Le, 2020) and ConvNext (Liu et al., 2022c) (CNN), and AST (Gong et al., 2021) (ViT). Additionally, we use two transformers for raw waveforms: EAT (Gazneli et al., 2022), focused on environmental sounds, and W2V2 (Baevski et al., 2020), known for speech processing. We utilize pre-trained weights based on their availability on HF. AST and EAT are pre-trained on AudioSet, corresponding to a transfer learning and fine-tuning approach in **DT**. Our training protocol also includes various augmentations to better align training and test data distributions: Time-shifting and background noise adjust for vocalization timings and integrate diverse noise profiles. Multi-label mixup (Zhang et al., 2018) increases sample diversity, and adding no-call training data from the auxiliary dataset VOX simulates non-vocal segments.

**Evaluation protocol.** We establish an evaluation protocol to assess model performance across `BirdSet`'s test datasets in various scenarios (e.g., class imbalance and geographical diversity), benchmarking a model's generalization capability and robustness in audio classification. We use threshold-free metrics to assess generalization without threshold-tuning, minimizing application-specific biases (Hendrycks & Gimpel, 2018). Following related work, we use cmAP (Kahl et al., 2023), AUROC (Van Merriënboer et al., 2024), and T1-Acc (Denton et al., 2022). This metric collection aligns with recent work in audio classification (Gemmeke et al., 2017; Chen et al., 2024) and supports evaluation across various use cases. We repeat the experiments across three (**MT**, **LT**) and five (**DT**) random seeds and report the mean results. Refer to Appendix C for more details and the standard deviations. Following (Hamer et al., 2023), we employ POW as a soundscape validation dataset for **LT** and **MT**. We save a checkpoint after each epoch and select the model with the lowest validation loss for testing. During inference, we restrict the models' logits by excluding those unrelated to the species present in the test datasets. Since validating with POW is not possible in **DT** as the model is limited to the dedicated classes, we allocate 20% of the training data for validation using the same checkpoint method. We also incorporate inference with Perch (Hamer et al., 2023) that has already shown promising results (Ghani et al., 2023) and exclude BirdNET (Kahl et al., 2021b) due to potential test data leakage.

### 4.2 EMPIRICAL RESULTS

Table 3 and Figure 6 report results for the **LT** scenario, detailing the generalization performance of the models in audio classification. We also present overall scores averaged across all test datasets for **DT** and **MT**. The results across all metrics and models indicate that large-scale representation learning in **LT** and **MT** generally improves performance compared to **DT**. We can see that fine-tuning pre-trained models from general audio domains (AST and W2V2) in dedicated environments in **DT** without domain-specific knowledge is less effective than employing large-scale models trained in **LT** or **MT**. Furthermore, models trained in the **LT** scenario demonstrate greater versatility, encompassing knowledge about approximately 10,000 bird species. Overall scores vary notably across datasets, underscoring the unique challenges in each test dataset. For instance, PER and UHH are challenging due to complex overlaps of multiple vocalizations and location-specific background noise, poorly represented in the focal training data (i.e., covariate shift). This variability emphasizes the importance of

diverse test data collections to effectively evaluate generalization capabilities in various environments. The T1-Acc metric indicates substantial difficulties in accurately classifying even one bird vocalizing in a segment. These results on `BirdSet` stress the necessity for ongoing research in environmental audio classification with domain-specific approaches to address the complexities of model robustness, particularly in managing label uncertainty, covariate shift, and task shift.

Table 3: Mean results across datasets and models for large training on XCL. **Best** and second best results are highlighted. Score reflects test average for all training scenarios, also including dedicated fine-tuning (**DT**) on subsets, medium training on XCM.

| | | Val | Test | | | | | | | Score | | |
|---|---|---|---|---|---|---|---|---|---|---|---|---|
| | | POW | PER | NES | UHH | HSN | NBP | SSW | SNE | LT | MT | DT |
| **Eff. Net** | cmAP | 0.35 | 0.17 | 0.30 | 0.23 | 0.35 | 0.57 | 0.33 | 0.28 | 0.32 | 0.36 | 0.35 |
| | AUROC | 0.82 | 0.71 | 0.88 | 0.78 | 0.86 | 0.90 | 0.91 | **0.83** | 0.84 | **0.85** | 0.84 |
| | T1-Acc | 0.80 | 0.38 | 0.49 | 0.42 | 0.59 | 0.63 | 0.55 | 0.67 | 0.53 | 0.59 | 0.54 |
| **Conv Next** | cmAP | 0.36 | **0.19** | 0.34 | 0.26 | **0.47** | 0.62 | **0.35** | **0.30** | 0.36 | 0.36 | **0.37** |
| | AUROC | 0.82 | **0.72** | 0.88 | **0.79** | **0.89** | **0.92** | **0.93** | **0.83** | **0.85** | 0.84 | 0.83 |
| | T1-Acc | 0.75 | 0.36 | 0.45 | 0.44 | 0.52 | 0.64 | 0.53 | 0.65 | 0.51 | 0.57 | 0.52 |
| **AST** | cmAP | 0.33 | 0.18 | 0.32 | 0.21 | 0.44 | 0.61 | 0.33 | 0.28 | 0.34 | 0.31 | 0.29 |
| | AUROC | 0.82 | **0.72** | 0.89 | 0.75 | 0.85 | 0.91 | **0.93** | 0.82 | 0.84 | 0.83 | 0.83 |
| | T1-Acc | 0.79 | 0.40 | 0.48 | 0.39 | 0.48 | 0.61 | 0.50 | 0.57 | 0.49 | 0.49 | 0.47 |
| **EAT** | cmAP | 0.27 | 0.12 | 0.27 | 0.22 | 0.38 | 0.50 | 0.25 | 0.24 | 0.30 | 0.33 | 0.33 |
| | AUROC | 0.79 | 0.64 | 0.87 | 0.76 | 0.86 | 0.87 | 0.90 | 0.81 | 0.80 | 0.82 | 0.78 |
| | T1-Acc | 0.69 | 0.32 | 0.46 | 0.40 | 0.47 | 0.61 | 0.46 | 0.58 | 0.48 | 0.47 | 0.47 |
| **W2V2** | cmAP | 0.27 | 0.14 | 0.30 | 0.21 | 0.40 | 0.57 | 0.29 | 0.25 | 0.31 | 0.29 | 0.26 |
| | AUROC | 0.75 | 0.68 | 0.86 | 0.76 | 0.86 | 0.90 | 0.90 | 0.78 | 0.78 | 0.80 | 0.79 |
| | T1-Acc | 0.72 | 0.34 | 0.47 | 0.51 | 0.50 | 0.65 | 0.50 | 0.51 | 0.50 | 0.46 | 0.44 |
| **Perch** | cmAP | 0.30 | 0.18 | **0.39** | **0.27** | 0.45 | **0.63** | 0.28 | 0.29 | 0.36 | - | - |
| | AUROC | 0.84 | 0.70 | **0.90** | 0.76 | 0.86 | 0.91 | 0.91 | **0.83** | 0.84 | - | - |
| | T1-Acc | 0.85 | **0.48** | **0.66** | **0.57** | 0.58 | **0.69** | **0.62** | **0.69** | **0.61** | - | - |

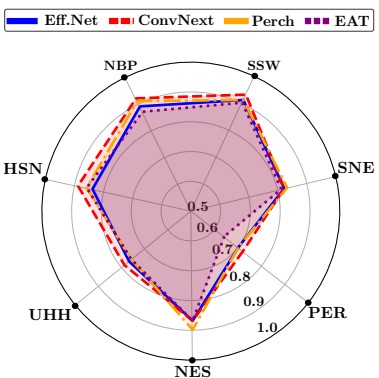

Figure 6: Mean AUROC in large training **LT** on XCL with selected models across test datasets.

The AUROC metric shows that our ConvNext implementation consistently outperforms other models across datasets in the **LT** scenario, especially in its ability to discriminate between positive bird vocalizations and negative classes or background noise. It often exceeds the SOTA model Perch (Hamer et al., 2023), which, despite its simpler EfficientNet architecture, excels in retrieval tasks as reflected by the T1-Acc metric. However, ConvNext's consistent performance, shown by the AUROC and cmAP metrics, highlights the advantages of more complex model architectures. We also see that complex models such as AST or ConvNext perform better than smaller ones such as EfficientNet in **LT**. This could guide future research towards prioritizing large embedding models for creating representations and fine-tuning in multi-label audio classification (Ghani et al., 2023). Waveform transformers such as EAT and W2V2 offer competitive yet slightly inferior performance compared to spectrogram-based models, possibly struggling to handle input noise.

## 5 CONCLUSION AND LIMITATIONS

**Conclusion.** We introduced `BirdSet`, a large-scale and multi-purpose audio classification dataset in avian bioacoustics. This versatile and challenging dataset offers a complex domain-specific case study for broader audio classification tasks. We identified and discussed key challenges in audio classification that are represented as evaluation use cases in `BirdSet`. As the primary evaluation use case, we benchmarked supervised multi-label classification under covariate shift, task shift, and label uncertainty using five well-known deep learning models across three unique training scenarios. These scenarios include large-scale supervised learning and fine-tuning on dedicated subsets. By providing these resources, we aim to set a new standard for classification tasks in passive acoustic monitoring. For future research, we aim to expand the benchmark by leveraging self-supervised representation learning techniques from the broader audio domain.

**Limitations.** While our benchmark indicates generalization performance in multi-label audio classification, it does not analyze `BirdSet`'s underlying characteristics affecting model performance, such as challenges associated with model robustness. Future research should focus on these aspects to enhance model robustness and applicability across diverse environments for audio classification. Additionally, our benchmark does not include the most recent audio transformers and a more thorough investigation of the influence of pre-training. However, our code is flexibly designed to support additional model implementations to extend our benchmark in the future.

## ETHICS STATEMENT

We collected and curated data under appropriate Creative Commons (CC) licenses to ensure privacy and uphold ethical standards. By facilitating passive recordings for testing in passive acoustic monitoring, we aim to minimize disturbance to natural habitats, promoting advancements in environmental audio classification and biodiversity monitoring. However, ethical and responsible use of both the dataset and the models derived from `BirdSet` is essential. The models developed from this dataset are strictly intended to advance audio research and biodiversity conservation. Each recording from XC is licensed and credited to the recordist. Researchers must respect these licenses, appropriately credit the recordist, and ensure their privacy. To promote comparability, accessibility, and reproducibility, we require all researchers using this dataset and benchmark to disclose their methodologies, report results openly, and state their research objectives. We aim to foster collaboration and advance the fields through this standardized dataset, encouraging ethical research practices and sharing results.

## REPRODUCIBILITY STATEMENT

Reproducibility is a primary goal of this work. To ensure this, we have made the dataset collection, `BirdSet`, openly and easily accessible via Hugging Face, along with all model checkpoints (Appendix A). The complete code for reproducing our benchmark results is available in a GitHub repository, which includes detailed instructions for setting up the environment, running the code, and following the training and evaluation protocols (Appendix A). These materials ensure that all experiments can be easily reproduced and extended. Further details regarding data collection, preprocessing, augmentations, evaluation, and hyperparameter configurations can be found in Appendix C

## ACKNOWLEDGEMENTS

This research was conducted under the DeepBirdDetect project (FKZ 67KI31040E), funded by the German Federal Ministry for the Environment, Nature Conservation, Nuclear Safety and Consumer Protection (BMUV).

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

APPENDIX

## A  OUTLINE

This supplementary material complements our main findings and provides additional information. Below, we outline the key resources included in our external materials.

---

**Overview Supplementary Material**

1. We have uploaded our large-scale **dataset collection** BirdSet[a] (Rauch et al., 2024b) to Hugging Face Datasets and provide an extensive **code repository**[b], which includes the training and evaluation protocols, is available on GitHub. We also provide a **leaderboard**[c] on Hugging Face to promote comparability and encourage participation in our benchmark.

2. To promote transparency and reproducibility, we have made all our **experimental results**[d] available through Weights&Biases (Biewald, 2020).

3. We provide access to our **trained models**[e] through Hugging Face Models (Wolf et al., 2020), allowing others to reproduce our results without the need to train from scratch. By doing so, we offer trainable baseline models to facilitate further experimentation and development.

4. We provide a **croissant** (Akhtar et al., 2024) metadata file in our code repository.

---

[a] https://huggingface.co/datasets/DBD-research-group/BirdSet
[b] https://github.com/DBD-research-group/BirdSet
[c] https://huggingface.co/spaces/DBD-research-group/BirdSet-Leaderboard
[d] https://wandb.ai/deepbirddetect/birdset/
[e] https://huggingface.co/DBD-research-group

---

The remainder of the supplementary material is structured as follows. Section B provides detailed insights into the BirdSet dataset collection. Section C offers comprehensive information on the experimental setup, including the metrics used and an in-depth presentation of the results. Section E presents a structured datasheet that offers a clear overview of the benchmark, facilitating a thorough understanding of its scope and structure.

## B  BIRDSET: A DATASET COLLECTION

This section provides additional details on the BirdSet dataset collection. We include information on licensing, maintenance, and hosting, data collection processes, dataset descriptions, and dataset statistics.

### B.1  LICENSING

The dataset collection BirdSet (Rauch et al., 2024b) is available under the creative commons (CC) CC-BY-NC-SA license. Researchers are permitted to use this dataset exclusively for non-commercial research and educational purposes. Each training recording in BirdSet sourced from XC is associated with its own CC license, which can be accessed via the metadata file on Hugging Face. We have excluded all recordings with non-derivative (ND) licenses. All test datasets are licensed under CC-BY-4.0, and the validation dataset is licensed under CC0-1.0.

Table 4: License overview in the training dataset from XC.

| License | XC recordings [#] |
|---|---|
| CC-BY-NC | 128 |
| CC-BY-SA | 7,293 |
| CC-BY | 199 |
| CC-BY-NC-SA | 484,350 |
| CC-0 | 706 |

We have carefully collected the recordings of this dataset collection. We, the authors, bear all responsibility to remove data or withdraw the paper in case of violating licensing or privacy rights upon confirmation of such violations. However, users are responsible for ensuring that their dataset use complies with all licenses, applicable laws, regulations, and ethical guidelines. We make no representations or warranties of any kind and accept no responsibility in the case of violations.

## B.2 MAINTENANCE AND HOSTING

The dataset is hosted on Hugging Face Höchst et al. (2022), enabling fast and easy accessibility. We maintain a complete backup of the dataset collection on our servers to ensure long-term availability. In addition to the hosted dataset, we provide a data loading script and respective metadata, allowing manual data retrieval. The dataset will be supported, hosted, and maintained by the DBD-research-group, initiated by the Deep Bird Detect project. The authors of this paper are committed to maintaining the code and dataset collection. For any inquiries or support requests, please contact the primary author at lukas.rauch@uni-kassel.de. Updates to the dataset will be implemented as necessary. While the test datasets will remain unchanged to ensure consistency in evaluations, the training datasets may be expanded or refined in response to new data that becomes available.

## B.3 DATA COLLECTION

The data collection process for the test and training datasets was planned to provide high-quality, representative, and legally compliant data. We carefully curated the test datasets from a range of high-quality soundscape recordings that simulate a real-world PAM environment. We chose these datasets to represent different difficulty levels, encompassing class diversity, class imbalance, geographical variations, and the density of vocalizations within each segment. Such diversity is essential for evaluating the robustness and generalizability of the models across diverse scenarios, ensuring the test datasets accurately reflect practical conditions. Additionally, the collection process was guided by the availability of appropriate licenses, guaranteeing that all included recordings are legally compliant and suitable for use on platforms such as Hugging Face.

Table 5 displays the metadata we provide for `BirdSet`. Note that not all information is available for every recording and recording type. For instance, detected events or peaks are included in the focal training dataset, not in the test datasets, since soundscape recordings are precisely annotated with specific timeframes for vocalizations. Additionally, for focal recordings, sources from XC, secondary calls, or the species' sex are only sometimes available.

Table 5: Metadata of the dataset collection `BirdSet` (Rauch et al., 2024b) on Hugging Face Datasets. We mark the availability of the respective information in train or test.

| | Format | Description | Train | Test |
|---|---|---|---|---|
| audio | Audio(sampling_rate=32_000) | audio recording from hf | ✓ | ✓ |
| filepath | Value("string") | relative path where the recording is stored | ✓ | ✓ |
| start_time | Value("float") | start time of a vocalization in s | ✗ | ✓ |
| end_time | Value("float") | end time of a vocalization in s | ✗ | ✓ |
| low_freq | Value("int") | low frequency bound for a vocalization in kHz | ✗ | ✓ |
| high_freq | Value("int") | high frequency bound for a vocalization in kHz | ✗ | ✓ |
| ebird_code | ClassLabel(names=class_list) | assigned species label | ✓ | ✓ |
| ebird_code_secondary | Sequence(datasets.Value("string")) | possible secondary species in a recording | ✓ | ✗ |
| ebird_code_multilabel | Sequence(datasets.ClassLabel(names=class_list)) | assigned species label in a multilabel format | ✓ | ✓ |
| call_type | Sequence(datasets.Value("string")) | type of bird vocalization | ✓ | ✗ |
| sex | Value("string") | sex of bird species | ✗ | ✓ |
| lat | Value("float") | latitude of vocalization/recording in WGS84 | ✓ | ✓ |
| long | Value("float") | longitude of vocalization/recording in WGS84 | ✓ | ✓ |
| length | Value("int") | length of the file in s | ✓ | ✓ |
| license | Value("string") | license of the recording | ✓ | ✓ |
| source | Value("string") | source of the recording | ✓ | ✓ |
| local_time | Value("string") | local time of the recording | ✓ | ✓ |
| detected_events | Sequence(datasets.Sequence(datasets.Value("float"))) | detected events in a recording with bambird, tuples of timestamps | ✓ | ✗ |
| event_cluster | Sequence(datasets.Value("int")) | detected audio events assigned to a cluster with bambird | ✓ | ✗ |
| peaks | Sequence(datasets.Value("float")) | peak event detected with scipy peak detection | ✓ | ✗ |
| quality | Value("string") | recording quality of the recording from XC (A,B,C) | ✓ | ✗ |
| recordist | Value("string") | recordist of the recording from XC | ✓ | ✗ |

## B.4 DATASETS OVERVIEW

Table 6: Avian data sources in related work.

| Abbr. | Reference | #Labels | Open-Source |
|---|---|---|---|
| BD | BirdsData | 14,311 | ✗ |
| NES | Colombia Costa Rica (Vega-Hidalgo et al., 2023) | 6.952 | ✓ |
| CAP | Caples Denton et al. (2022) | 2,944 | ✗ |
| CBI | Cornell Bird Identification Howard et al. (2020) | 21,382 | ✓ |
| HSN | High Sierra Nevada Clapp et al. (2023) | 21,375 | ✓ |
| INA | INaturalist Chasmai et al. (2024) | 604,284 | ✓ |
| MAC | Macaulay Library | 14,530 | ✗ |
| NBP | NIPS4BPlus Morfi et al. (2018) | 1,687 | ✓ |
| PER | Amazon Basin Hopping et al. (2022) | 14,798 | ✓ |
| POW | Powdermill Nature Chronister et al. (2021) | 16,052 | ✓ |
| S2L | Soundscapes2Landscapes Clark et al. (2023) | - | ✗ |
| SNE | Sierra Nevada Kahl et al. (2022c) | 20,147 | ✓ |
| SSW | Sapsucker Woods Kahl et al. (2022a) | 50,760 | ✓ |
| UHH | Hawaiian Islands Navine et al. (2022) | 59,583 | ✓ |
| VOX | BirdVox Lostanlen et al. (2018a) | 35,402 | ✓ |
| XC | Xeno-CantoVellinga & Planqué (2015) | 677,429 | ✓ |

**Amazon Basin (PER).** The Amazon Basin (Hopping et al., 2022) **soundscape test dataset** comprises 21 hours of annotated audio recordings from the Southwestern Amazon Basin, featuring 14,798 bounding box labels for 132 bird species. These recordings were collected at the Inkaterra Reserva Amazonica in Peru during early 2019. The dataset includes diverse forest conditions and was partly used for the 2020 BirdCLEF competition (Kahl et al., 2020). Annotations specifically targeted the dawn hour across seven sites on multiple dates to capture the peak vocal activity of neotropical birds.

**Columbia Costa Rica (NES).** The Columbia Costa Rica (Vega-Hidalgo et al., 2023) **soundscape test dataset** contains 34 hours of annotated audio recordings from landscapes of Colombian and Costa Rican coffee farms. This dataset comprises 6,952 labels for 89 bird species, and was partially featured in the 2021 BirdCLEF competition (Kahl et al., 2021a). The recordings primarily capture dawn choruses from randomly sampled farm locations and dates.

**Hawaiian Islands (UHH).** The Hawaiian Islands (Navine et al., 2022) **soundscape test dataset** features 635 recordings totaling nearly 51 hours, annotated by ornithologists with 59,583 labels for 27 bird species. This dataset includes recordings from endangered native species from Hawaii, gathered across four sites on Hawai'i Island from 2016 to 2022. It was utilized in the 2022 BirdCLEF competition (Kahl et al., 2022b).

**High Sierra Nevada (HSN).** The High Sierras (Clapp et al., 2023) **soundscape test dataset** comprises 100 ten-minute audio recordings from the southern Sierra Nevada in California, annotated with over 10,000 labels for 21 bird species. This dataset was featured in the 2020 BirdCLEF competition (Kahl et al., 2020). The recordings were captured in high-elevation regions of Sequoia and Kings Canyon National Parks to study the ecological impact of trout on birds. Annotations were rigorously verified by experts.

**NIPS4Bplus (NBP).** The Neural Information Processing Scaled for Bioacoustics Plus (Morfi et al., 2018) is an enhanced version of the dataset NIPS4B 2013, developed for bird song classification with detailed temporal species annotations. Collected across diverse climates and geographical features in France and Spain, this **soundscape test dataset** was initially targeted at bat echolocation but also successfully captured bird songs. It is designed to maximize species diversity, distinguishing it from other European datasets regarding the number of recordings and species classes. This dataset undergoes more extensive processing to achieve a balance among classes, simplifying the classification task compared to other datasets.

**Powdermill Nature (POW).** The Powdermill Nature (Chronister et al., 2021) **soundscape validation dataset** is a collection of strongly-labeled bird soundscapes from Powdermill Nature Reserve in the Northeastern United States. It features 385 minutes of dawn chorus recordings, captured over four

days from April to July 2018. These recordings include vocalizations from 48 bird species, with a total of 16,052 labels. Since it includes species that overlap with other test datasets and can be processed quickly, it has been selected as a validation dataset.

**Sapsucker Woods (SSW).** The Sapsucker Woods **soundscape test dataset** (Kahl et al., 2022a) consists of 285 hour-long soundscape recordings from the Sapsucker Woods bird sanctuary in Ithaca, New York. Captured in 2017 by the Cornell Lab of Ornithology, these high-quality audio recordings include 50,760 annotations for 81 bird species. The dataset primarily aims to study vocal activity patterns, local bird species' seasonal diversity, and noise pollution's impact on birds. It has also been instrumental in the 2019, 2020, and 2021 BirdCLEF (Kahl et al., 2020; 2021a; 2022b) competitions. The annotation process involved carefully selecting and labeling bird calls, with each call boxed in time and frequency.

**Sierra Nevada (SNE)** (Kahl et al., 2022c) The Sierra Nevada **soundscape test dataset** features 33 hour-long audio recordings from the Sierra Nevada region in California, annotated with 20,147 labels for 56 bird species. These recordings, collected in 2018, were specifically gathered to study the impact of forest management on avian populations. The dataset was part of the 2021 BirdCLEF (Kahl et al., 2021a) competition. It was captured across diverse ecological settings in the Lassen and Plumas National Forests, targeting various elevations and latitudes.

**BirdVox-DCASE-20k (VOX).** Under the VOX abbreviation, we compile **soundscape auxiliary dataset** derived from the DCASE18 bird audio detection challenge (Stowell et al., 2019). This challenge required participants to design systems capable of detecting the presence of birds in a binary classification setting using soundscape recordings. As these training datasets have ground truth labels regarding the presence of bird sounds, they are well-suited for background augmentations for segment-based PAM. It incorporates 20,000 audio clips from remote monitoring units in Ithaca, NY, USA, focusing on flight calls of birds (Lostanlen et al., 2018b).

**Xeno-Canto (XC) (Vellinga & Planqué, 2015)** XC sources all **focal training datasets** and acts as a citizen-science platform dedicated to avian sound recording. XC was launched in 2005 and aims to popularize bird sound recording, improve accessibility to bird sounds, and expand knowledge in this field. As a collaborative repository, it allows enthusiasts worldwide to upload bird sounds with essential metadata, including species, the recorder's name, location, and date. The platform allows anyone with internet access to upload recordings, provided they include a minimum set of metadata such as species, recordist name, location, and recording date. Despite varying submission quality, all contributions are valuable, capturing rare or common bird vocalizations. XC's collection features nearly 10,00 bird species with more than 800,000 recordings, and its collection is continuously growing.

### B.5 DATA COLLECTION STATISTICS

We explore the characteristics of the `BirdSet` dataset collection, emphasizing the soundscape test and validation datasets. We display the species composition of test and validation datasets in Figure 7. Additionally, we highlight the overlaps between test and validation segments in Figure 8. Figure 9 shows the number of annotations per species in the XCL and evaluation datasets. This analysis not only clarifies the datasets' structures but also aids in understanding the varying difficulty levels of the test datasets, reflecting the influence of different environmental and biological factors. Table 7 complements our graphical abstract in the introduction, showcasing a range of datasets in environmental audio classification.

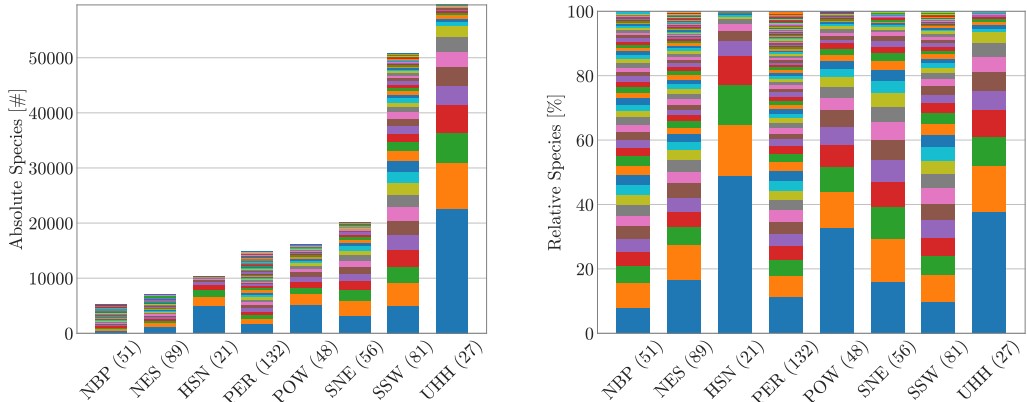

Figure 7: Species composition of test and validation datasets, presented in absolute counts (#) and relative percentages (%). Colored sections indicate unique species within each dataset. Identical colors do not correspond to the same species.

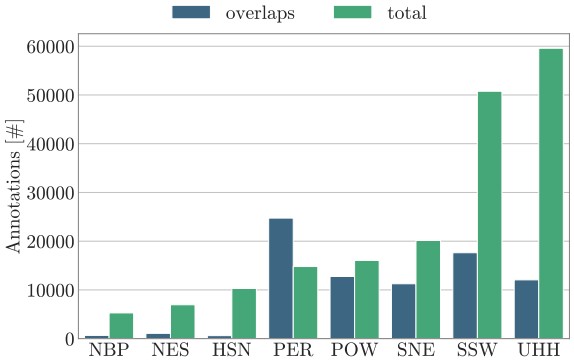

Figure 8: The number of annotations where overlaps are present. Annotations indicate the number of annotations present in each dataset.

Table 7: Available datasets in environmental audio classification.

| Dataset | Clips | Length | Duration [h] | Classes | Source | Domain | Label Level |
|---|---|---|---|---|---|---|---|
| BirdSet (train) | 528k | <1h | 6,877 | 10,296 | Xeno-Canto | Avian | File |
| BirdSet (eval) | 322k | 5s | 411 | 503 | Xeno-Canto | Avian | Event |
| iNatSounds (train) Chasmai et al. (2024) | 137k | | 1,100 | 5,569 | iNaturalist | Avian | File |
| iNatSounds (eval) Chasmai et al. (2024) | 95k | | 451 | 1,212 | iNaturalist | Avian | File |
| AudioSet (train) Gemmeke et al. (2017) | 1.8M | ≈10s | 5,790 | 527 | YouTube | Universal | Event |
| AudioSet (eval) Gemmeke et al. (2017) | 300k | ≈10s | 56 | 527 | YouTube | Universal | File |
| FSD Fonseca et al. (2017) | 297k | | 628 | 632 | Freesound | Universal | File |
| FSD50K (train) Fonseca et al. (2021) | 51k | 0.3-30s | 108 | 200 | Freesound | Universal | File |
| FSD50K (eval) Fonseca et al. (2021) | 10k | 0.3-30s | 28 | 200 | Freesound | Universal | File |
| VGG (train) Chen et al. (2020) | 200k | 10s | 550 | 310 | YouTube | Universal | File |
| VGG (eval) Chen et al. (2020) | 21k | 10s | 43 | 310 | YouTube | Universal | File |
| SONYC-UST-V2 (train) Cartwright et al. (2020) | 14k | 10s | 37 | 30 | SONYC | Urban | File |
| SONYC-UST-V2 (eval) Cartwright et al. (2020) | 5k | 10s | 14 | 30 | SONYC | Urban | File |
| ESC50 Piczak (2015) | 2k | 5s | 3 | 50 | Freesound | Environmental | Event |

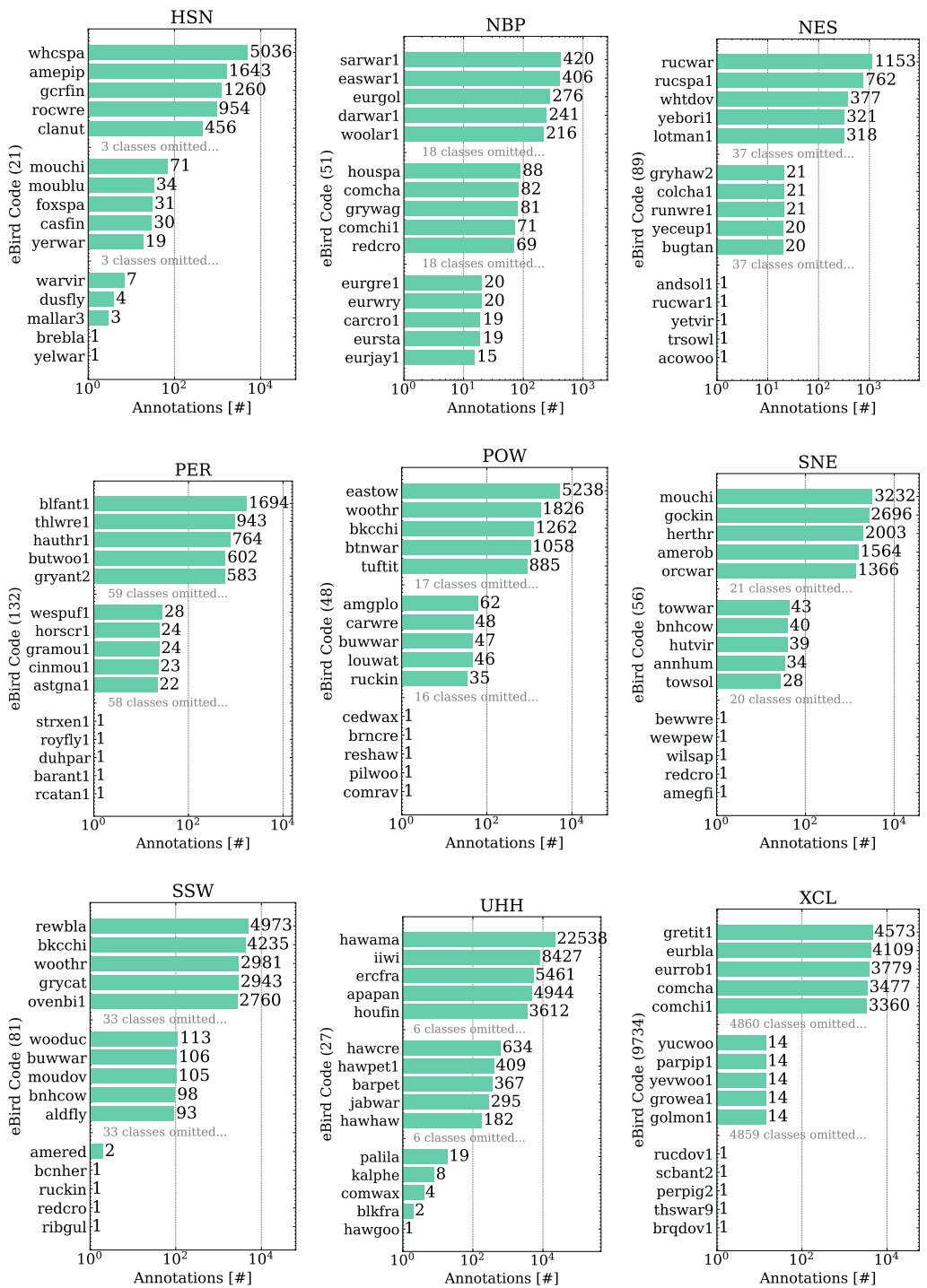

Figure 9: Top, middle, and bottom five number of species appearances (vocalizations) in each evaluation and the complete XCL training dataset.

## C  DETAILED EXPERIMENTAL SETTING AND RESULTS

This section offers additional experimental insights, detailing the computational resources and assets utilized. It outlines the experimental setup, including model parameters, describes the augmentations applied, and provides specifics about the evaluation metrics. We also include a comprehensive presentation of the results.

### C.1  COMPUTATIONAL RESOURCES

We primarily utilized an internal Slurm cluster equipped with NVIDIA A100 and V100 GPU servers from the IES group at the University of Kassel, predominantly using the NVIDIA A100 GPU servers for large-scale training. Collaboration with researchers from the University of Kiel and Fraunhofer IEE also involved using NVIDIA A100 GPUs within their internal compute clusters. Additionally, we conducted smaller-scale experiments on a workstation equipped with an NVIDIA RTX 4090 GPU and an AMD Ryzen 9 7950X CPU. Due to variations in the conditions across different clusters, direct comparisons of training times are challenging. However, we provide detailed training and inference times and additional details for our computational resources on Weights&Biases.

### C.2  ASSETS

In our benchmark and code, we leverage a variety of existing assets. By integrating these existing assets, we can focus on our core research objectives by utilizing well-established tools and platforms. Our primary assets include:

- *Hugging Face Datasets* (Höchst et al., 2022) (platform and code, Apache-2.0 license): We use the Hugging Face Datasets platform for hosting and processing our data collection. This enables fast and efficient accessibility.

- *Hugging Face Models* (Wolf et al., 2020) (platform and code, Apache-2.0 license): For model deployment, we utilize various models from Hugging Face Models, including EfficientNet (Tan & Le, 2020), AST (Gong et al., 2021), Wav2Vec2 (Baevski et al., 2020), and ConvNext (Liu et al., 2022c). Hugging Face is also utilized to host the checkpoints of our trained models.

- *PyTorch* (Paszke et al., 2019) (code) and *PyTorch Lightning* (Falcon & The PyTorch Lightning team, 2019) (code, Apache-2.0 license): Our codebase is built using PyTorch and the PyTorch Lightning frameworks. This provides a robust and scalable environment for developing and training our models.

- *Torch Audiomentations* (Jordal et al., 2024) (code, MIT license): We employ Torch Audiomentations for data augmentation, allowing us to enhance the variability and robustness of our training data.

- *BamBird (Michaud et al., 2023)* (code, BSD-3-Clause license): BamBird is used for detecting events in the focal recordings, facilitating training and efficient analysis.

- *Hydra* (Yadan, 2019) (code, MIT license): We use Hydra for experiment management, enabling us to organize and streamline our experimental workflows effectively.

- *Weights&Biases* (Biewald, 2020) (platform and code, MIT license): We utilize Weights&Biases for tracking our experiments and publishing our results in detail.

- *Zenodo* (CERN) (platform) and *Xeno-Canto* (Vellinga & Planqué, 2015) (platform) were the source of the train, test, and validation dataset collection.

These assets enable us to establish a readily accessible dataset collection, fostering reproducibility and comparability of our results.

### C.3  EXPERIMENTAL SETUP

Table 8 provides a detailed overview of the parameters used to generate baselines for all training scenarios. It serves as an addition to the main article.

Table 8: Model and training parameters for training scenarios LT and MT and DT*. Note that Perch is not further trained in our benchmark and can only be employed as a black box for inference. The species limit caps the number of samples from any one species to prevent imbalance, while the event limit restricts extractions per recording to ensure dataset diversity in the training data.

| Parameter | **Perch** (Hamer et al., 2023) | **EfficientNet** (Tan & Le, 2020) | **ConvNext** (Liu et al., 2022c) | **AST** (Gong et al., 2021) | **EAT** (Gazneli et al., 2022) | **W2V2** (Baevski et al., 2020) |
|---|---|---|---|---|---|---|
| Model parameters | | | | | | |
| Input type | Spec | Spec | Spec | Spec | Wave | Wave |
| Pretrained | - | ImageNet | ImageNet | AudioSet | - | LibriSpeech |
| Epochs | 1M steps | 30 | 30 | 12 | 30 | 40 |
| Optimizer | Adam | AdamW | AdamW | AdamW | AdamW | AdamW |
| Weight decay | None | 5e-4 | 5e-4 | 1e-2 | 1e-5 | 1e-2 |
| Learning rate | 1e-3 | 5e-4 | 5e-4 | 1e-5 | 3e-4 | 3e-4 |
| Scheduler | - | Cos.-Anneal. | Cos.-Anneal. | Cos.-Anneal. | Cos.-Anneal. | Cos.-Anneal. |
| Warmup ratio | None | 5e-2 | 5e-2 | 5e-2 | 5e-2 | 5e-2 |
| Batch Size | 256 | 64 | 64 | 12 | 128 | 64 |
| Validation | POW | POW, 20%Tr* | POW, 20%Tr* | POW, 20%Tr* | POW, 20%Tr* | POW, 20%Tr* |
| Loss | BCE | BCE | BCE | BCE | BCE | BCE |
| Architecture | B1 | B1 | Base224 | - | S | Base |
| # Parameters | 19.0 M | 19.0 M | 97.5 M | 93.7 M | 5.2 M | 97.1 M |
| Spectrogram parameters | | | | | | |
| # fft | 1024 | 2048 | 1024 | 1024 | ✗ | ✗ |
| hop length | 320 | 256 | 320 | 320 | ✗ | ✗ |
| power | 2.0 | 2.0 | 2.0 | 2.0 | ✗ | ✗ |
| Melscale parameters | | | | | | |
| # Mels | 160 | 256 | 128 | 128 | ✗ | ✗ |
| # STFT | - | 1025 | 513 | 513 | ✗ | ✗ |
| DB Scale | PCEN | ✓ | ✓ | ✓ | ✗ | ✗ |
| Processing parameters | | | | | | |
| Norm wave | peak norm | ✗ | ✗ | instance norm | instance norm | intstance norm |
| Norm spec | ✗ | AudioSet M/Std | AudioSet M/Std | AudioSet M/Std | ✗ | ✗ |
| Resize | ✗ | ✗ | ✗ | 1024 | ✗ | ✗ |
| # Seeds | 1 | 3,5* | 3,5* | 3,5* | 3,5* | 3,5* |
| Data parameters | | | | | | |
| Sampling rate | 32kHz | 32 kHz | 32 kHz | 32 kHz | 32 kHz | 16 kHz |
| Event limit | 5 | 1, 0, 5* | 1 0, 5* | 10, 5* | 1 0, 5* | 10, 5* |
| Species limit | ✗ | 500, 0, 600* | 500, 0, 600* | 500, 0, 600* | 500, 0, 600* | 500, 0, 600* |
| # Train samples | 750,000 | 1,528,068, variable, 558,455* | | | | |

## C.4 AUGMENTATIONS

To counteract covariate shift, we aim to pinpoint augmentations that effectively bridge the gap between the training distribution (focal recordings) and the test distribution (soundscape recordings). Moreover, we seek to tackle the task shift dilemma when transitioning from a multi-class to a multi-label setting. We differentiate between waveform augmentations and spectrogram augmentations:

**Waveform augmentations** are applied before a spectrogram conversion on the raw waveform. They are always used independently of the input type.

- **Time Shifting** ($p=1.0$) modifies the temporal aspect of the audio to enhance model robustness against timing variations. Upon detecting a vocalization event, we capture an 8-second window surrounding it and then randomly select a 5-second segment from this window. This approach ensures that vocalizations are not confined to the start of an event.

- **Background Noise Mixing** ($p=0.5$) incorporates diverse noise profiles to train the model in distinguishing signal from noise. This augmentation is particularly beneficial as it allows us to incorporate the VOX auxiliary dataset, which consists of background noise from soundscape recordings without bird vocalizations. It plays a crucial role in enhancing the evaluation performance.

- **Gain Adjustments** ($p=0.2$) vary the volume to ensure the model's resilience to amplitude fluctuations. Additionally, this method assists in aligning the training's focal recordings closer to the characteristics of soundscape recordings, which are captured with omnidirectional microphones featuring diverse spatial attributes.

- **Multi-Label Mixup** ($p=0.7$) enhances sample diversity by mixing up to two samples from a batch, extending the conventional Mixup (Hendrycks et al., 2020) technique to include the labels of the combined recordings. This adaptation aims to boost model generalization and

address task shifts, effectively creating a synthetic multi-label challenge within a multi-class dataset. Moreover, it narrows the gap between the train and test distributions.

- **No-Call Mixing** ($p$=0.075) enriches training by substituting samples with segments labeled as no-call (0-vector). This adjustment is crucial for segment-based evaluation in PAM scenarios, where soundscapes may contain segments devoid of bird vocalizations. This situation is not encountered in training focal recordings, where events are pre-identified by a recordist. We also leverage the VOX auxiliary dataset, which comprises soundscapes devoid of any sounds, to simulate no-call scenarios effectively.

**Spectrogram augmentations** are applied after the spectrogram conversion. They cannot be used when the input type only accepts raw waveforms.

- **Frequency Masking** ($p$=0.5) enhances robustness to variations in spectral components by randomly obscuring a contiguous range of frequency bands in the spectrogram. This approach aims to simulate real-world PAM scenarios where certain frequency components might be masked due to environmental factors or recording anomalies. By introducing this variation during training, frequency masking helps the model to be less sensitive to specific frequency absences, promoting better generalization across diverse acoustic conditions.
- **Time Masking** ($p$=0.3) improves resilience to temporal variations by obscuring a segment of time in the spectrogram. This approach mimics interruptions or fluctuations in sound continuity that often occur in PAM settings, such as noises or overlapping sounds.

### C.5 EVALUATION

We opt for threshold-free metrics to obtain a clear view of overall model performance without the necessity of fine-tuning thresholds for individual classes. This approach enhances comparability and minimizes biases toward particular applications. The metrics implemented are the following:

- **cmAP** (class mean average precision) computes the AP (average precision) for each class $c$ as an element independently and then averages these scores across all classes $C$, resulting in a macro average (Kahl et al., 2023; Denton et al., 2022; Höchst et al., 2022).

$$\text{cmAP} := \frac{1}{C} \sum_{c=1}^{C} \text{AP}(c) \tag{1}$$

This metric reflects the model's ability to rank positive instances higher than negative ones across all possible threshold levels, offering a comprehensive view of model performance without threshold calibration. However, it is noisy for species with sparse labels, limiting comparability across datasets (Denton et al., 2022).

- **Top-1 Accuracy** evaluates whether the class with the highest predicted probability is (one of) the correct class for each instance:

$$\text{T1-Acc} := \frac{1}{N} \sum_{i=1}^{N} \mathbf{1}[\hat{y}_i \in Y_i], \tag{2}$$

where $N$ represents the total number of instances in the dataset, $\hat{y}_i$ denotes the class with the highest predicted probability and $Y_i$ is the set of true labels. The indicator function $\mathbf{1}[\hat{y}_i \in Y_i]$ outputs 1 if the predicted class $\hat{y}_i$ is one of the true labels in $Y_i$, and 0 otherwise. Although not a traditional multi-label metric, it offers an easily interpretable measure of generalization performance, particularly useful in practical scenarios where identifying a bird species is the primary concern.

- **AUROC** (area under the receiver operating characteristic curve) computes the area under the receiver operating characteristic curve given a model $f$:

$$\text{AUROC}(f) := \frac{\sum_{x_0 \in D^0} \sum_{x_1 \in D^1} [f(x_0) < f(x_1)]}{|D^0| \cdot |D^1|}, \tag{3}$$

where $D^0$ is a set of negative and $D^1$ positive examples ("Calders & Jaroszewicz, 2007). It summarizes the curve into one number, measuring the model's ability to distinguish

between classes across all thresholds. It is threshold-independent and provides a balanced performance view, especially in class imbalance cases with an expected value of 0.5 for each random ranking (Hamer et al., 2023).

In addition to individually reporting the results for both training scenarios, we compute the aggregate performance score averaged across the complete benchmark. We aim to facilitate a quick and comprehensive comparison between models.

## C.6 VALIDATION

Due to the extensive empirical experiments and the volume of the training datasets required to produce our baseline results for our benchmark within the `BirdSet` collection, traditional hyperparameter optimization was not feasible. Additionally, the covariate shift between focals in training and soundscapes in testing complicates result validation. Utilizing a validation split from the training data is only partially functional, as it does not accurately represent the soundscape data distribution. Furthermore, obtaining soundscapes from the test dataset for validation could introduce test data leakage due to the static nature of soundscape recordings. Therefore, we follow (Hamer et al., 2023) and employ the soundscape dataset POW for validation. However, this approach only allows us to validate the results of the MT and LT training scenarios since DT has dedicated classes that are not available in the validation dataset. In this case, we used a validation split from the training data and employed augmentations.

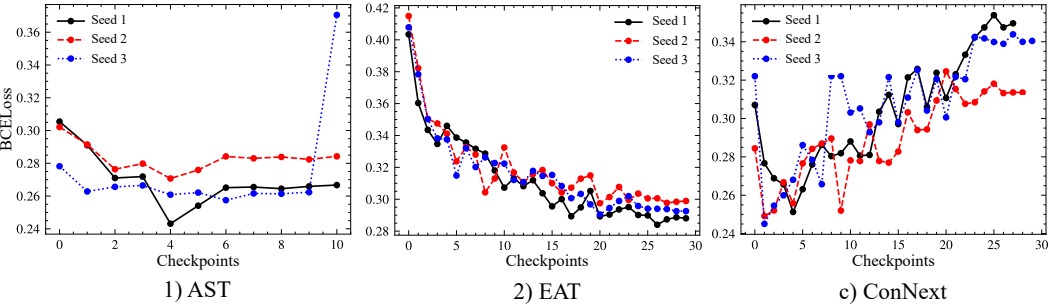

Figure 10: Validation loss in the scenario LT on POW with the models AST, EAT, and ConvNext.

We used the validation dataset to select the best-performing model based on the lowest loss across all training epochs on POW. We show exemplary results of the validation loss curves for the training scenario LT in Figure 10. It illustrates that more complex models with extensive parameters (AST and ConvNext) tend to overfit early on the focal training data. In contrast, the smaller model (EAT) benefits from extended training duration. More detailed results are available on Weights&Biases.

## C.7 DETAILED BENCHMARK RESULTS

We present detailed results for our three training scenarios in `BirdSet`. DT represents dedicated training on the respective subsets (see Table 9 and Figure 11). MT indicates medium training on approximately 400 species found in the test datasets (see Table 10 and Figure 12). LT denotes large training on nearly 10,000 bird species from XC (see Table 11 and Figure 13).

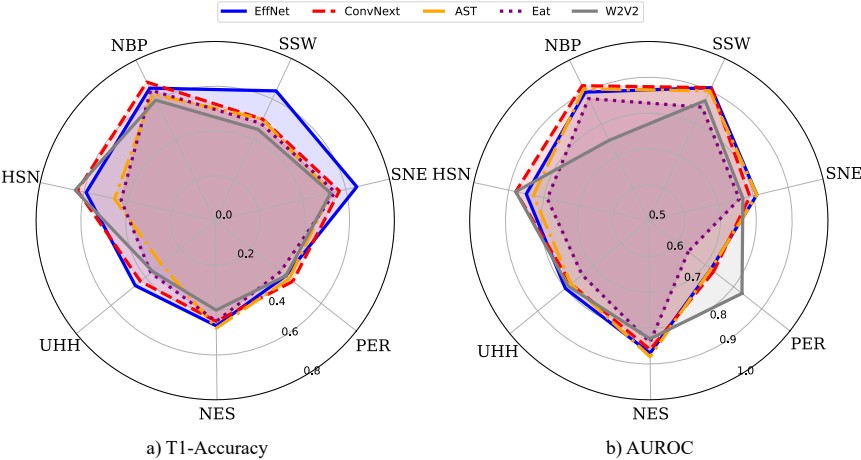

Figure 11: Selected results for DT.

Table 9: Mean results and standard deviations across 5 seeds in scenario **dedicated training** (DT).

| | | POW$^v$ | PER | NES | UHH | HSN | NBP | SSW | SNE | Score |
|---|---|---|---|---|---|---|---|---|---|---|
| **Eff. Net** | cmAP | $0.39_{\pm 0.03}$ | $\underline{0.19}_{\pm 0.01}$ | $\mathbf{0.33}_{\pm 0.00}$ | $\mathbf{0.24}_{\pm 0.01}$ | $\underline{0.44}_{\pm 0.02}$ | $\underline{0.64}_{\pm 0.02}$ | $\mathbf{0.35}_{\pm 0.01}$ | $\mathbf{0.28}_{\pm 0.01}$ | $\underline{0.35}$ |
| | AUROC | $0.83_{\pm 0.01}$ | $\underline{0.72}_{\pm 0.01}$ | $\underline{0.87}_{\pm 0.01}$ | $\mathbf{0.80}_{\pm 0.03}$ | $\underline{0.85}_{\pm 0.02}$ | $0.90_{\pm 0.02}$ | $\mathbf{0.91}_{\pm 0.01}$ | $\mathbf{0.81}_{\pm 0.01}$ | $\mathbf{0.84}$ |
| | T1-Acc | $0.70_{\pm 0.02}$ | $0.40_{\pm 0.02}$ | $\underline{0.47}_{\pm 0.01}$ | $\mathbf{0.46}_{\pm 0.01}$ | $\underline{0.59}_{\pm 0.02}$ | $\underline{0.66}_{\pm 0.02}$ | $\mathbf{0.54}_{\pm 0.01}$ | $\mathbf{0.65}_{\pm 0.01}$ | $\mathbf{0.54}$ |
| **Conv Next** | cmAP | $0.38_{\pm 0.01}$ | $\mathbf{0.21}_{\pm 0.01}$ | $\underline{0.32}_{\pm 0.01}$ | $\underline{0.23}_{\pm 0.01}$ | $\mathbf{0.46}_{\pm 0.02}$ | $\mathbf{0.67}_{\pm 0.01}$ | $\underline{0.33}_{\pm 0.01}$ | $\underline{0.26}_{\pm 0.02}$ | $\underline{0.37}$ |
| | AUROC | $0.83_{\pm 0.01}$ | $\mathbf{0.73}_{\pm 0.01}$ | $0.86_{\pm 0.01}$ | $\underline{0.78}_{\pm 0.01}$ | $\mathbf{0.88}_{\pm 0.02}$ | $\mathbf{0.92}_{\pm 0.00}$ | $\mathbf{0.91}_{\pm 0.01}$ | $\underline{0.79}_{\pm 0.02}$ | $\underline{0.83}$ |
| | T1-Acc | $0.67_{\pm 0.01}$ | $\mathbf{0.44}_{\pm 0.01}$ | $0.45_{\pm 0.02}$ | $\underline{0.43}_{\pm 0.01}$ | $\mathbf{0.62}_{\pm 0.02}$ | $\mathbf{0.69}_{\pm 0.01}$ | $\underline{0.50}_{\pm 0.02}$ | $\underline{0.57}_{\pm 0.02}$ | $\underline{0.52}$ |
| **AST** | cmAP | $0.32_{\pm 0.01}$ | $\underline{0.19}_{\pm 0.02}$ | $0.29_{\pm 0.01}$ | $0.15_{\pm 0.01}$ | $0.30_{\pm 0.01}$ | $0.59_{\pm 0.01}$ | $0.30_{\pm 0.01}$ | $0.24_{\pm 0.01}$ | $0.29$ |
| | AUROC | $0.82_{\pm 0.01}$ | $\underline{0.72}_{\pm 0.01}$ | $\mathbf{0.88}_{\pm 0.01}$ | $\underline{0.78}_{\pm 0.02}$ | $0.83_{\pm 0.02}$ | $\underline{0.91}_{\pm 0.00}$ | $\underline{0.90}_{\pm 0.01}$ | $\mathbf{0.81}_{\pm 0.01}$ | $\mathbf{0.83}$ |
| | T1-Acc | $0.69_{\pm 0.01}$ | $\underline{0.42}_{\pm 0.01}$ | $\mathbf{0.48}_{\pm 0.01}$ | $0.31_{\pm 0.01}$ | $0.46_{\pm 0.02}$ | $0.63_{\pm 0.02}$ | $\underline{0.50}_{\pm 0.02}$ | $0.53_{\pm 0.01}$ | $0.48$ |
| **EAT** | cmAP | $0.32_{\pm 0.01}$ | $0.14_{\pm 0.01}$ | $0.29_{\pm 0.01}$ | $0.17_{\pm 0.00}$ | $0.35_{\pm 0.02}$ | $0.57_{\pm 0.01}$ | $0.27_{\pm 0.01}$ | $0.23_{\pm 0.01}$ | $0.33$ |
| | AUROC | $0.80_{\pm 0.00}$ | $0.64_{\pm 0.01}$ | $0.84_{\pm 0.00}$ | $0.74_{\pm 0.01}$ | $0.79_{\pm 0.01}$ | $0.88_{\pm 0.00}$ | $0.85_{\pm 0.01}$ | $0.76_{\pm 0.00}$ | $0.78$ |
| | T1-Acc | $0.73_{\pm 0.01}$ | $0.37_{\pm 0.01}$ | $0.45_{\pm 0.01}$ | $0.37_{\pm 0.01}$ | $0.43_{\pm 0.01}$ | $0.65_{\pm 0.01}$ | $0.48_{\pm 0.01}$ | $0.55_{\pm 0.02}$ | $0.47$ |
| **W2V2** | cmAP | $0.24_{\pm 0.02}$ | $0.11_{\pm 0.10}$ | $0.24_{\pm 0.01}$ | $0.15_{\pm 0.01}$ | $0.34_{\pm 0.01}$ | $0.54_{\pm 0.02}$ | $0.22_{\pm 0.01}$ | $0.21_{\pm 0.01}$ | $0.26$ |
| | AUROC | $0.75_{\pm 0.02}$ | $0.64_{\pm 0.01}$ | $0.83_{\pm 0.01}$ | $0.73_{\pm 0.01}$ | $0.79_{\pm 0.02}$ | $0.88_{\pm 0.01}$ | $0.87_{\pm 0.01}$ | $0.77_{\pm 0.00}$ | $0.79$ |
| | T1-Acc | $0.61_{\pm 0.03}$ | $0.30_{\pm 0.02}$ | $0.40_{\pm 0.02}$ | $0.38_{\pm 0.01}$ | $0.36_{\pm 0.03}$ | $0.64_{\pm 0.03}$ | $0.45_{\pm 0.02}$ | $0.53_{\pm 0.06}$ | $0.44$ |

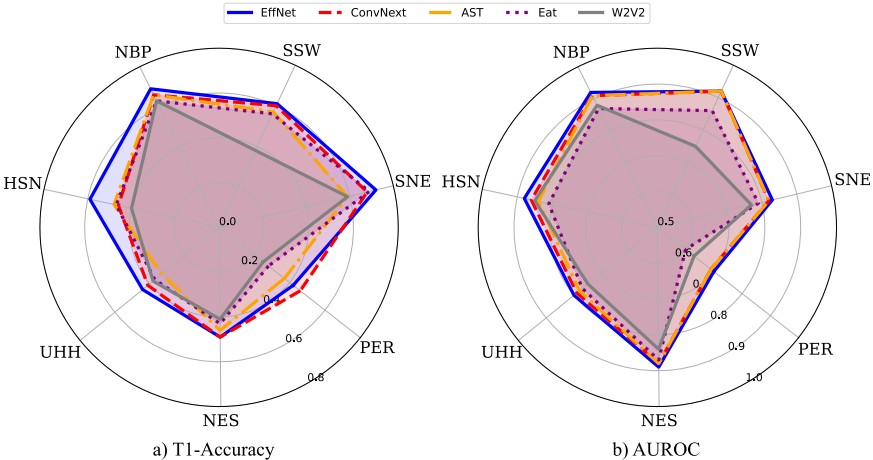

Figure 12: Selected results for MT.

Table 10: Mean results and standard deviations across 3 seeds in scenario **medium training** (MT).

| | | POW | PER | NES | UHH | HSN | NBP | SSW | SNE | Score |
|---|---|---|---|---|---|---|---|---|---|---|
| **Eff. Net** | cmAP | 0.42± 0.01 | **0.18**± 0.00 | **0.32**± 0.01 | **0.26**± 0.02 | **0.48**± 0.01 | **0.64**± 0.01 | **0.36**± 0.01 | **0.30**± 0.01 | **0.36** |
| | AUROC | 0.85± 0.00 | **0.70**± 0.00 | **0.89**± 0.01 | **0.80**± 0.02 | **0.88**± 0.02 | **0.92**± 0.00 | **0.92**± 0.01 | **0.83**± 0.01 | **0.85** |
| | T1-Acc | 0.79± 0.04 | 0.42± 0.02 | **0.49**± 0.01 | **0.44**± 0.05 | **0.59**± 0.03 | **0.69**± 0.01 | **0.61**± 0.04 | **0.72**± 0.03 | **0.59** |
| **Conv Next** | cmAP | 0.40± 0.04 | **0.18**± 0.01 | **0.32**± 0.01 | 0.24± 0.02 | 0.47± 0.01 | **0.64**± 0.03 | **0.36**± 0.02 | 0.28± 0.01 | **0.36** |
| | AUROC | 0.84± 0.01 | 0.69± 0.01 | 0.88± 0.01 | 0.79± 0.02 | 0.86± 0.02 | 0.91± 0.01 | **0.92**± 0.01 | 0.82± 0.01 | 0.84 |
| | T1-Acc | 0.79± 0.04 | **0.46**± 0.04 | **0.49**± 0.01 | 0.41± 0.09 | 0.46± 0.08 | 0.66± 0.02 | 0.60± 0.04 | 0.68± 0.04 | 0.57 |
| **AST** | cmAP | 0.33± 0.00 | 0.15± 0.01 | 0.29± 0.01 | 0.19± 0.00 | 0.38± 0.02 | 0.61± 0.02 | 0.30± 0.01 | 0.26± 0.01 | 0.31 |
| | AUROC | 0.82± 0.00 | 0.69± 0.01 | 0.88± 0.01 | 0.78± 0.01 | 0.84± 0.03 | 0.91± 0.01 | 0.90± 0.01 | 0.82± 0.01 | 0.83 |
| | T1-Acc | 0.75± 0.00 | 0.37± 0.03 | 0.46± 0.01 | 0.33± 0.01 | 0.48± 0.05 | 0.66± 0.03 | 0.57± 0.02 | 0.59± 0.01 | 0.49 |
| **EAT** | cmAP | 0.29± 0.01 | 0.09± 0.00 | 0.28± 0.01 | 0.21± 0.01 | 0.40± 0.01 | 0.54± 0.02 | 0.27± 0.01 | 0.23± 0.01 | 0.29 |
| | AUROC | 0.78± 0.01 | 0.60± 0.00 | 0.87± 0.00 | 0.77± 0.02 | 0.81± 0.01 | 0.87± 0.01 | 0.86± 0.00 | 0.79± 0.02 | 0.80 |
| | T1-Acc | 0.76± 0.00 | 0.27± 0.01 | 0.43± 0.01 | 0.37± 0.01 | 0.47± 0.04 | 0.63± 0.04 | 0.56± 0.01 | 0.64± 0.02 | 0.48 |
| **W2V2** | cmAP | 0.26± 0.02 | 0.10± 0.01 | 0.27± 0.02 | 0.19± 0.01 | 0.41± 0.02 | 0.56± 0.05 | 0.26± 0.01 | 0.21± 0.02 | 0.29 |
| | AUROC | 0.76± 0.01 | 0.63± 0.03 | 0.84± 0.00 | 0.75± 0.02 | 0.85± 0.03 | 0.88± 0.02 | 0.85± 0.01 | 0.77± 0.02 | 0.80 |
| | T1-Acc | 0.76± 0.02 | 0.25± 0.01 | 0.41± 0.02 | 0.38± 0.05 | 0.40± 0.04 | 0.63± 0.04 | 0.55± 0.02 | 0.59± 0.04 | 0.46 |

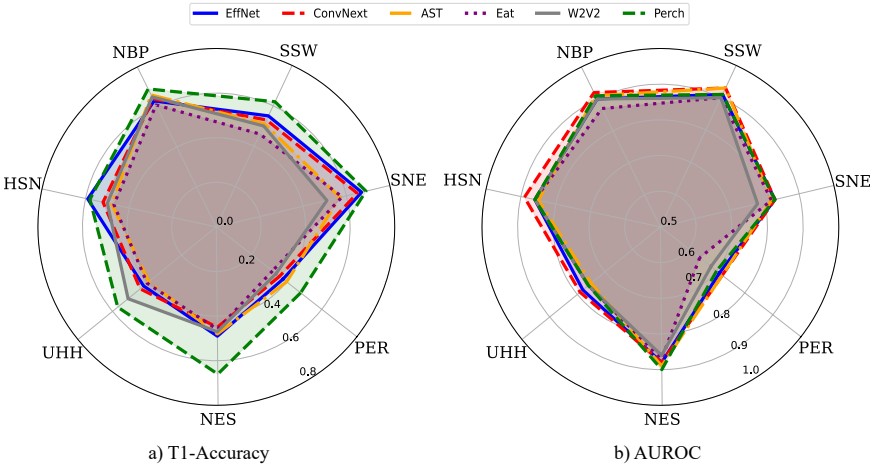

Figure 13: Selected results for LT.

Table 11: Mean results and standard deviations across 5 seeds in scenario **large training** (LT).

| | | POW | PER | NES | UHH | HSN | NBP | SSW | SNE | Score |
|---|---|---|---|---|---|---|---|---|---|---|
| **Eff. Net** | cmAP | 0.35± 0.01 | 0.17± 0.02 | 0.30± 0.03 | 0.23± 0.01 | 0.35± 0.15 | 0.57± 0.06 | 0.33± 0.03 | 0.28± 0.03 | 0.32 |
| | AUROC | 0.82± 0.01 | 0.71± 0.02 | 0.88± 0.02 | 0.78± 0.02 | 0.86± 0.04 | 0.90± 0.02 | 0.91± 0.02 | **0.83± 0.02** | 0.84 |
| | T1-Acc | 0.80± 0.03 | 0.38± 0.01 | 0.49± 0.01 | 0.42± 0.03 | **0.59± 0.05** | 0.63± 0.04 | 0.55± 0.02 | 0.67± 0.03 | 0.53 |
| **Conv Next** | cmAP | 0.36± 0.01 | **0.19± 0.01** | 0.34± 0.01 | 0.26± 0.01 | **0.47± 0.02** | 0.62± 0.02 | **0.35± 0.01** | **0.30± 0.01** | 0.36 |
| | AUROC | 0.82± 0.02 | **0.72± 0.01** | 0.88± 0.01 | **0.79± 0.02** | **0.89± 0.01** | **0.92± 0.01** | **0.93± 0.00** | **0.83± 0.01** | **0.85** |
| | T1-Acc | 0.75± 0.05 | 0.36± 0.07 | 0.45± 0.01 | 0.44± 0.03 | 0.52± 0.01 | 0.64± 0.00 | 0.53± 0.03 | 0.65± 0.02 | 0.51 |
| **AST** | cmAP | 0.33± 0.01 | 0.18± 0.01 | 0.32± 0.01 | 0.21± 0.01 | 0.44± 0.02 | 0.61± 0.02 | 0.33± 0.01 | 0.28± 0.01 | 0.34 |
| | AUROC | 0.82± 0.00 | **0.72± 0.01** | 0.89± 0.01 | 0.75± 0.01 | 0.85± 0.02 | 0.91± 0.00 | **0.93± 0.01** | 0.82± 0.01 | 0.84 |
| | T1-Acc | 0.79± 0.04 | 0.40± 0.02 | 0.48± 0.00 | 0.39± 0.01 | 0.48± 0.02 | 0.66± 0.03 | 0.51± 0.01 | 0.57± 0.04 | 0.49 |
| **EAT** | cmAP | 0.27± 0.01 | 0.12± 0.00 | 0.27± 0.00 | 0.22± 0.00 | 0.38± 0.01 | 0.50± 0.00 | 0.25± 0.00 | 0.24± 0.00 | 0.30 |
| | AUROC | 0.79± 0.01 | 0.64± 0.01 | 0.87± 0.01 | 0.76± 0.01 | 0.86± 0.01 | 0.87± 0.00 | 0.90± 0.00 | 0.82± 0.00 | 0.82 |
| | T1-Acc | 0.69± 0.02 | 0.32± 0.01 | 0.46± 0.02 | 0.40± 0.02 | 0.47± 0.02 | 0.61± 0.01 | 0.46± 0.02 | 0.58± 0.02 | 0.48 |
| **W2V2** | cmAP | 0.27± 0.04 | 0.14± 0.00 | 0.30± 0.01 | 0.21± 0.01 | 0.40± 0.02 | 0.57± 0.03 | 0.29± 0.01 | 0.25± 0.01 | 0.31 |
| | AUROC | 0.75± 0.01 | 0.68± 0.00 | 0.86± 0.01 | 0.76± 0.04 | 0.86± 0.00 | 0.90± 0.01 | 0.90± 0.00 | 0.78± 0.03 | 0.78 |
| | T1-Acc | 0.72± 0.02 | 0.34± 0.06 | 0.47± 0.01 | 0.51± 0.03 | 0.50± 0.05 | 0.65± 0.06 | 0.50± 0.00 | 0.51± 0.03 | 0.50 |
| **Perch** | cmAP | 0.30 | 0.18 | **0.39** | **0.27** | 0.45 | **0.63** | 0.28 | 0.29 | 0.36 |
| | AUROC | 0.84 | 0.70 | **0.90** | 0.76 | 0.86 | 0.91 | 0.91 | 0.83 | 0.84 |
| | T1-Acc | 0.85 | **0.48** | **0.66** | 0.57 | 0.58 | **0.69** | 0.62 | 0.69 | **0.61** |

## D    INFERENCE RESULTS

In this section, we exemplify inference results on HSN and POW for our ConvNext model and Perch, both pretrained on XC, in a threshold-dependent setting. We adopt the same configuration as BirdNET (Kahl et al., 2021b), utilizing a global prediction threshold of 0.1.

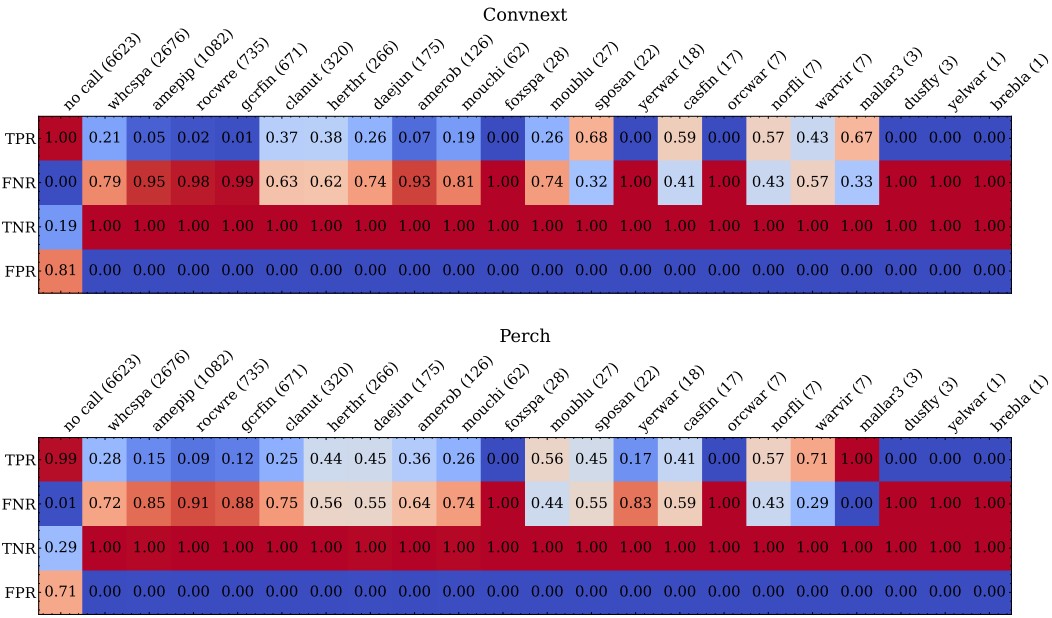

Figure 14: **HSN** Classification performance in **LT** showing True Positive Rate (TPR), False Negative Rate (FNR), True Negative Rate (TNR), and False Positive Rate (FPR). The special case of *no call* where the segment has no annotation and the model predicts a 0-vector. The X-axis label shows each class's occurrences (not necessarily equal to the #Annotations).

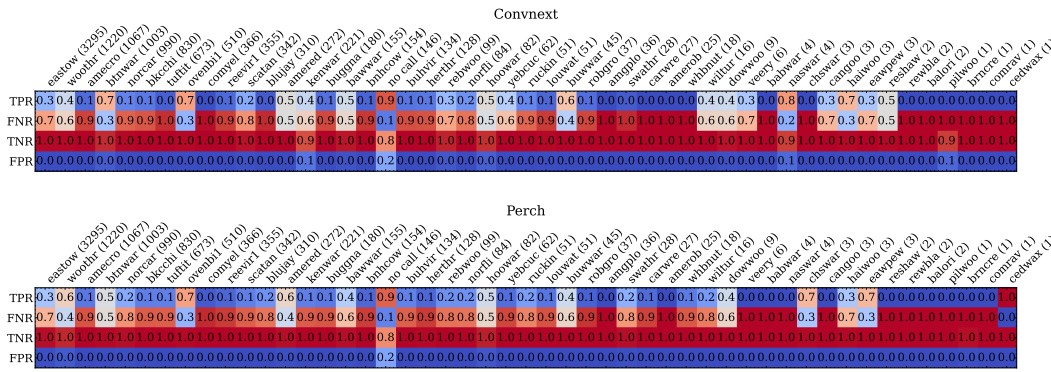

Figure 15: **POW** Classification performance in **LT** showing True Positive Rate (TPR), False Negative Rate (FNR), True Negative Rate (TNR), and False Positive Rate (FPR). The special case of *no call* where the segment has no annotation and the model predicts a 0-vector. The X-axis label shows each class's occurrences (not necessarily equal to the #Annotations).

# E  DATASHEET

This section provides a comprehensive and structured datasheet `BirdSet`, following the guidelines outlined by Gebru et al. (2021). We aim to enhance clarity and facilitate effective communication between us and researchers who wish to utilize the dataset collection. The datasheet comprises the dataset's motivation, composition, collection process, preprocessing steps, uses, distribution, and maintenance.

## E.1  MOTIVATION

**For what purpose was the dataset created?** *Was there a specific task in mind? Was there a specific gap that needed to be filled? Please provide a description.*

The `BirdSet` dataset collection was created to address the demand for a large-scale audio representation learning dataset, including a real-world evaluation test set. In addition, `BirdSet` serves as a standardized dataset and evaluation protocol in computational avian bioacoustics. The specific task is the classification of bird vocalizations in a realistic passive acoustic monitoring (PAM) scenario. The goal was to consolidate fragmented research efforts, improve reproducibility, comparability, and accessibility, and ultimately enhance the effectiveness of deep learning (DL) models in analyzing bird vocalizations.

**Who created the dataset (e.g., which team, research group) and on behalf of which entity (e.g., company, institution, organization)?** *If there is an associated grant, please provide the name of the grantor and the grant name and number.*

The dataset collection was created by a collaborative effort of researchers from the University of Kassel at the Intelligent Embedded Systems group (IES), Kiel University at the Intelligent Systems group (INS), Fraunhofer Institute for Energy Economics and Energy System Technology (IEE), and TU Chemnitz. Throughout the creation of the dataset, the following people have been involved: Lukas Rauch (IES), Raphael Schwinger (CAU), Moritz Wirth (IEE), and René Heinrich (IEE).

**Who funded the creation of the dataset?** The research was funded by the German Ministry for Environment, Nature Conservation, Nuclear Safety, and Consumer Protection through the project "DeepBirdDetect - Automatic Bird Detection of Endangered Species Using Deep Neural Networks" (grant numbers: 67KI31040A, 67KI31040B, 67KI31040C).

**Any other comments?** A more detailed motivation and corresponding objectives can be found in the main article.

## E.2  COMPOSITION

**What do the instances that comprise the dataset represent (e.g., documents, photos, people, countries)?** *Are there multiple types of instances (e.g., movies, users, and ratings; people and interactions between them; nodes and edges)? Please provide a description.*

The instances in the dataset collection are comprised of audio recordings of bird vocalizations in their natural habitats. These include focal recordings from Xeno-Canto (XC), actively recorded by a recordist for training purposes, and soundscape recordings captured by omnidirectional microphones and annotated by experts for testing. For more detailed information on the unified metadata provided for each instance, refer to Table 5.

**How many instances are there in total (of each type, if appropriate)?**

The dataset contains a total of 712,515 instances. It includes over 520,000 focal recordings from various locations worldwide and over 400 hours of soundscape recordings from passive acoustic monitoring (PAM) scenarios in South America, North America, and Europe. For more details, refer to Figure 7, Figure 8, Figure 5, and the main article.

**Does the dataset contain all possible instances, or is it a sample (not necessarily random) of instances from a larger set? If the dataset is a sample, then what is the larger set? Is the sample representative of the larger set (e.g., geographic coverage)?** *If so, please describe how this representativeness was validated/verified. If it is not representative of the larger set, please*

*describe why not (e.g., to cover a more diverse range of instances because instances were withheld or unavailable)*

The dataset is a curated collection of publicly available recordings, primarily sourced from the platforms XC (Vellinga & Planqué, 2015) and Zenodo (CERN). We included all available recordings from XC as of 03/10/2024 in the XCL dataset, excluding those with an ND license. Additionally, we omitted recordings of highly endangered bird species. The XCM dataset, a subset of XCL, contains all bird species present across the test dataset. We also provide dedicated training datasets containing only the species represented in each test dataset. Selected test datasets from Zenodo are incorporated as available. Further details can be found in the main article.

**What data does each instance consist of?** *"Raw" data (e.g., unprocessed text or images) or features? In either case, please provide a description*

An instance consists of a raw audio recording with a resolution of 32 kHz in the `.OGG` format and respective metadata detailed in Table 5. We also provide a list of possible vocalization events for each training recording.

**Is there a label or target associated with each instance?** *If so, please provide a description.*

Each instance is labeled with the eBird taxonomy (Sullivan et al., 2009) from 2021 corresponding to the bird species in the recording.

**Is any information missing from individual instances?** *If so, please provide a description, explaining why this information is missing (e.g., because it was unavailable). This does not include intentionally removed information but might include, e.g., redacted text.*

No information from the source datasets has been removed. However, some recordings in the training datasets have incomplete metadata and annotations. While secondary bird vocalizations are sometimes annotated, they are not always identified. Additionally, it may not always be clear which events belong to the primary bird species or secondary background species. More information on label uncertainty is provided in the main article.

**Are relationships between individual instances made explicit (e.g., users' movie ratings, social network links)?** *If so, please describe how these relationships are made explicit.*

Relationships between instances are not explicitly defined, as each recording is treated independently for the classification tasks. However, the metadata includes details such as the recordist, location, and recording device, allowing for identifying similar instances. The soundscape test datasets are recorded in static environments where segments have inherent relationships that are valuable for real-world applications. These relationships are not considered in our benchmark.

**Are there recommended data splits (e.g., training, development/validation, testing)?** *If so, please provide a description of these splits, explaining the rationale behind them.*

The dataset collection is split into training and testing datasets. Training datasets (XCL, XCM) are used to train large-scale models, while soundscape recordings (e.g., PER, NES, UHH) are used for testing. This split reflects practical PAM scenarios, where models are trained on a broad dataset and evaluated on realistic, strongly-labeled soundscapes. Additionally, we provide a validation dataset (POW) for tuning hyperparameters or validating model results. Additionally, we provide small fine-tuning training datasets for each test dataset with a randomly generated validation split during training.

**Are there any errors, sources of noise, or redundancies in the dataset?** *If so, please provide a description.*

Potential noise sources in the recordings include background sounds inherent to the natural environments where the data were collected. These are not typical errors but reflect the real-world conditions under which bird vocalizations occur. Additionally, the training recordings from XC are weakly labeled, meaning the exact timing of the vocalization events within the recordings is uncertain. Further details on label uncertainty are provided in the main paper.

**Is the dataset self-contained, or does it link to or otherwise rely on external resources (e.g., websites, tweets, other datasets)?** *If it links to or relies on external resources, a) are there guarantees that they will exist, and remain constant, over time; b) are there official archival versions of the*

*complete dataset (i.e., including the external resources as they existed at the time the dataset was created); c) are there any restrictions (e.g., licenses, fees) associated with any of the external resources that might apply to a dataset consumer? Please provide descriptions of all external resources and any restrictions associated with them, as well as links or other access points, as appropriate.*

The dataset is self-contained, with an internal backup ensuring its permanence and consistency over time. Given the dynamic nature of XC and the availability of additional data, the training datasets are designed to be expandable. Moreover, the metadata for each instance includes links to the original sources, facilitating easy access and verification.

**Does the dataset contain data that might be considered confidential (e.g., data that is protected by legal privilege or by doctor-patient confidentiality, data that includes the content of individuals' non-public communications)?** *If so, please provide a description.*

All metadata adheres to the licensing terms of the respective recordings on XC (including the recordists' names) and Zenodo.

**Does the dataset contain data that, if viewed directly, might be offensive, insulting, threatening, or might otherwise cause anxiety?**

No, the dataset collection only contains audio recordings of bird vocalizations, which are not offensive or threatening.

**Does the dataset identify any subpopulations (e.g., by age, gender)***If so, please describe how these subpopulations are identified and provide a description of their respective distributions within the dataset.*

The dataset does not identify subpopulations based on demographic factors; it focuses solely on bird species.

**Is it possible to identify individuals (i.e., one or more natural persons), either directly or indirectly (i.e., in combination with other data) from the dataset?** *If so, please describe how*

The metadata includes the recordist's name and the recording location, which may indirectly provide information about the recordist. The respective licenses of the XC recordings also include these details.

**Does the dataset contain data that might be considered sensitive in any way (e.g., data that reveals race or ethnic origins, sexual orientations, religious beliefs, political opinions or union memberships, or locations; financial or health data; biometric or genetic data; forms of government identification, such as social security numbers; criminal history)?** *If so, please provide a description.*

No, the dataset does not contain sensitive data.

**Any other comments?**

No further comments.

### E.3 COLLECTION PROCESS

**How was the data associated with each instance acquired?** *Was the data directly observable (e.g., raw text, movie ratings), reported by subjects (e.g., survey responses), or indirectly inferred/derived from other data (e.g., part-of-speech tags, model-based guesses for age or language)? If the data was reported by subjects or indirectly inferred/derived from other data, was the data validated/verified? If so, please describe how.*

The data was downloaded from the XC and Zenodo platforms. Download and preprocessing scripts are available in the GitHub repository.

**What mechanisms or procedures were used to collect the data** *(e.g., hardware apparatuses or sensors, manual human curation, software programs, software APIs)? How were these mechanisms or procedures validated?*

We collected the data via the XC API and Zenodo. The recordings were captured using equipment suitable for recording bird vocalizations. The recordings were manually curated and annotated by

experts or enthusiasts. For further details, see the descriptions of the individual datasets listed in Section B.

**If the dataset is a sample from a larger set, what was the sampling strategy** *(e.g., deterministic, probabilistic with specific sampling probabilities)?*

The dataset is a curated collection from larger collections, specifically from the open-source platforms XC and Zenodo. We exclude all recordings from XC that are licensed as CC-ND and sample based on our training scenarios; the snapshot date is 03/10/2024. The selected datasets from Zenodo are utilized as provided. See section B and the main article for further information.

**Who was involved in the data collection process** *(e.g., students, crowd workers, contractors) and how were they compensated (e.g., how much were crowd workers paid)?*

The data collection involved researchers from several institutions, including the University of Kassel, Kiel University, Fraunhofer Institute, and TU Chemnitz. The researchers were compensated according to their contracts. The annotators of the focal recordings freely uploaded their recordings and annotations to XC under respective licenses.

**Over what timeframe was the data collected?** *Does this timeframe match the creation timeframe of the data associated with the instances (e.g., recent crawl of new news articles)? If not, please describe the timeframe in which the data associated with the instances was created.*

The recordings were collected over several years on XC, with the date of each recording included in the metadata. A snapshot from XC was taken on 03/10/2024.

**Did you collect the data from the individuals in question directly or obtain it via third parties or other sources (e.g., websites)?**

The data was obtained from third-party sources, primarily the open-source platforms XC for training and Zenodo for testing datasets.

**Were the individuals in question notified about the data collection?** *If so, please describe (or show with screenshots or other information) how notice was provided, and provide a link or other access point to, or otherwise reproduce, the exact language of the notification itself.*

The dataset collection includes only recordings that are properly licensed for use. All recordings are under a creative commons (CC) license, summarized by our collection with the license CC-BY-NC-SA.

**Did the individuals in question consent to the collection and use of their data?** *If so, please describe (or show with screenshots or other information) how consent was requested and provided, and provide a link or other access point to, or otherwise reproduce, the exact language to which the individuals consented.*

We included only recordings with a proper CC license that permits usage.

**If consent was obtained, were the consenting individuals provided with a mechanism to revoke their consent in the future or for certain uses?** *If so, please provide a description, as well as a link or other access point to the mechanism (if appropriate).*

See above.

**Has an analysis of the potential impact of the dataset and its use on data subjects (e.g., a data protection impact analysis) been conducted?** *If so, please provide a description of this analysis, including the outcomes, as well as a link or other access point to any supporting documentation.*

No formal analysis has been conducted. However, all the data has already been publicly available before.

**Any other comments?**

No further comments.

### E.4    Preprocessing/Cleaning/Labeling

**Was any preprocessing/cleaning/labeling of the data done (e.g., discretization or bucketing, tokenization, part-of-speech tagging, SIFT feature extraction, removal of instances, processing of missing values)?** *If so, please provide a description. If not, you may skip the remaining questions in this section.*

The audio recordings have been converted to the .OGG format and standardized to a 32kHz resolution. Metadata has been standardized, and scientific bird species names have been unified to eBird (Sullivan et al., 2009) codes as labels. Refer to Table 5 for a comprehensive overview of the metadata. Additionally, we identified potential bird vocalization events from XC using the *bambird* (Michaud et al., 2023) event detection algorithm and *scipy peak detection*. The soundscape recordings have also been processed and are provided in 5-second segments for evaluation. For more details, refer to our Hugging Face Dataset collection.

**Was the "raw" data saved in addition to the preprocessed/cleaned/labeled data (e.g., to support unanticipated future uses)?** *If so, please provide a link or other access point to the "raw" data.*

The raw recordings remain accessible at their original sources. We also maintain an internal server backup of all data for added security and reliability.

**Is the software that was used to preprocess/clean/label the data available?** *If so, please provide a link or other access point.*

Code used for processing is available at our GitHub repository. Additionally, we detail the utilized software assets above.

**Any other comments?**

No further comments.

### E.5    Uses

**Has the dataset been used for any tasks already?** *If so, please provide a description.*

Yes, the dataset has been used in multi-label experiments in a PAM scenario with different training protocols outlined in the main paper. Additionally, the dataset has been used for other internal experiments in the context of the research project "DeepBirdDetect".

**Is there a repository that links to any or all papers or systems that use the dataset?** *If so, please provide a link or other access point.*

To date, no papers have utilized this dataset.

**What (other) tasks could the dataset be used for?**

The `BirdSet` dataset collection can be used for various tasks beyond multi-label classification, including but not limited to:

- Audio representation learning,
- Audio event detection,
- Noisy label learning,
- Few-Shot Learning,
- Domain adaption under covariate shift,
- Active learning,
- Conducting conservation research by monitoring bird populations and behaviors.

**Is there anything about the composition of the dataset or the way it was collected and pre-processed/cleaned/labeled that might impact future uses?** *For example, is there anything that a dataset consumer might need to know to avoid uses that could result in unfair treatment of individuals or groups (e.g., stereotyping, quality of service issues) or other risks or harms (e.g., legal risks, financial harms)? If so, please provide a description. Is there anything a dataset consumer could do to mitigate these risks or harms?*

The dataset is composed of audio recordings of bird vocalizations and related metadata. Since it does not involve human subjects, there are no risks of unfair treatment of individuals or groups. However, users should be aware of the inherent variability in recording conditions (e.g., background noise, distance from the sound source) and the potential for label noise due to the challenges in accurately annotating bird calls in natural environments.

**Are there tasks for which the dataset should not be used?** *If so, please provide a description.* The dataset should not be used to train models that can identify critically endangered bird species.

**Any other comments?**

No further comments.

### E.6 DISTRIBUTION

**Will the dataset be distributed to third parties outside of the entity (e.g., company, institution, organization) on behalf of which the dataset was created?** *If so, please provide a description.*

Yes, see the next question.

**How will the dataset be distributed (e.g., tarball on website, API, GitHub)?** *Does the dataset have a digital object identifier (DOI)?*

The dataset is available on Hugging Face with a respective DOI. Additionally, the metadata, including links to the sources, is available in our GitHub repository, allowing for dataset construction without using Hugging Face.

**When will the dataset be distributed?**

It is already public.

**Will the dataset be distributed under a copyright or other intellectual property (IP) license, and/or under applicable terms of use (ToU)?** *If so, please describe this license and/or ToU, and provide a link or other access point to, or otherwise reproduce, any relevant licensing terms or ToU, as well as any fees associated with these restrictions.*

The dataset collection `BirdSet` is available under the CC-BY-NC-SA license. Each recording in the training datasets retains the corresponding license from XC. We excluded all recordings from XC under a CC-ND license. All soundscape recordings from the different datasets collected from Zenodo are licensed under CC-BY-4.0.

**Have any third parties imposed IP-based or other restrictions on the data associated with the instances?** *If so, please describe these restrictions, and provide a link or other access point to, or otherwise reproduce, any relevant licensing terms, as well as any fees associated with these restrictions.*

No third parties have imposed IP-based or other restrictions on the data. The dataset is sourced from publicly accessible platforms that support open-access principles.

**Do any export controls or other regulatory restrictions apply to the dataset or to individual instances?** *If so, please describe these restrictions, and provide a link or other access point to, or otherwise reproduce, any supporting documentation.*

No export controls or other regulatory restrictions apply to the dataset or its instances.

**Any other comments?**

No further comments.

### E.7 MAINTENANCE

**How can the owner/curator/manager of the dataset be contacted (e.g., email address)?**

The dataset can be managed and inquiries addressed through the GitHub repository or by contacting the main authors involved in its creation via their institutional email addresses provided below:

- Lukas Rauch: lukas.rauch@uni-kassel.de

- Raphael Schwinger: rsc@informatik.uni-kiel.de
- Moritz Wirth: moritz.wirth@iee.fraunhofer.de
- René Heinrich: rene.heinrich@iee.fraunhofer.de

**Is there an erratum?** *If so, please provide a link or other access point.*

No erratum has been discovered so far.

**Will the dataset be updated (e.g., to correct labeling errors, add new instances, delete instances)?** *If so, please describe how often, by whom, and how updates will be communicated to dataset consumers (e.g., mailing list, GitHub)?*

Updates to the dataset have not yet been planned to ensure the comparability of the benchmark. However, advancements in areas such as event detection could lead to the addition of vocalization events.

**If the dataset relates to people, are there applicable limits on the retention of the data associated with the instances (e.g., were the individuals in question told that their data would be retained for a fixed period of time and then deleted)?** *If so, please describe these limits and explain how they will be enforced.*

The data does not directly relate to people.

**Will older versions of the dataset continue to be supported/hosted/maintained?** *If so, please describe how. If not, please describe how its obsolescence will be communicated to dataset consumers.*

Dataset versions are provided through Hugging Face.

**If others want to extend/augment/build on/contribute to the dataset, is there a mechanism for them to do so?** *If so, please provide a description. Will these contributions be validated/verified? If so, please describe how. If not, why not? Is there a process for communicating/distributing these contributions to dataset consumers? If so, please provide a description.*

Yes, contributions from the community are welcome. Interested parties can submit their contributions via pull requests on the GitHub or Hugging Face repository. The dataset maintainers will review and validate all contributions before being accepted and integrated. Contributions and updates will be communicated to users through the repository's update log.

**Any other comments?**

No further comments.

