# OpenReview forum: "BirdSet: A Large-Scale Dataset for Audio Classification in Avian Bioacoustics"
_ICLR.cc/2025/Conference — ICLR 2025 Spotlight_

### Official Review · Reviewer_qFwz · 2024-10-27

**Soundness:** 4
**Presentation:** 4
**Contribution:** 4
**Rating:** 8
**Confidence:** 5

**Summary:**

The paper introduces BirdSet, a large-scale audio classification dataset for avian bioacoustics, with around 520,000 recordings from nearly 10,000 bird species. It includes over 6,800 hours of training data and 400 hours of evaluation data from diverse regions. The dataset supports tasks such as multi-label classification and self-supervised learning, with standardized training and evaluation protocols. A comprehensive literature review identifies key challenges in bioacoustics and provides research guidelines. The paper benchmarks multiple deep learning models and offers a codebase for reproducibility.

**Strengths:**

1. The introduction of *BirdSet* fills a notable gap in audio classification by providing a large-scale, domain-specific dataset for avian bioacoustics.
2. The paper is well-structured and clearly articulates the challenges in avian bioacoustics and audio classification more broadly.
3. The paper provides a thorough empirical evaluation using six well-known deep learning models, covering various training scenarios, including large-scale training and fine-tuning.

**Weaknesses:**

1. The literature review in Section 2 is quite extensive, occupying a significant portion of the main paper. While the analysis is comprehensive, it might be more beneficial to allocate more space to the dataset description and details, rather than using five pages for the related work.
2. The results focus mainly on the multi-label classification benchmark, with limited exploration of other use cases. Given the dataset’s availability of precise event timestamps, it would be valuable to include benchmarks for sound event detection. Additionally, providing results for other mentioned use cases would strengthen the paper.

**Questions:**

Please refer to the weaknesses above for the questions.

---

> ### Author Response · Authors · 2024-11-17
> **Authors' Answer: Dataset Description and Evaluation Use Cases**
>
> Dear reviewer, thank you for your valuable review and very positive feedback. We are particularly pleased that you recognize the gap that BirdSet addresses and appreciate the extensive empirical results. Below, we directly respond to your two main remarks and outline how we plan to incorporate your feedback:
>
> ### **(1) Allocate more space to dataset description**
>
> > The literature review in Section 2 is quite extensive, occupying a significant portion of the main paper. While the analysis is comprehensive, it might be more beneficial to allocate more space to the dataset description and details, rather than using five pages for the related work.
>
> *Explanation and statement:* Our primary motivation for the extensive Section 2 was to introduce the broader ML/DL community to this field and present related work within a clear and structured context. While we acknowledge that Section 2 occupies quite a large portion of the paper, a comprehensive overview of the unique challenges is crucial to engaging a wider audience of ML/DL researchers. By highlighting the potential of this field for a range of ML/DL use cases, we aim to foster greater interest and involvement. We needed to strike a balance between providing essential dataset details and explaining the broader issues within this domain. Given that many readers may be unfamiliar with these challenges, we prioritized a beginner-friendly introduction. At the same time, exhaustive dataset details are available in the appendix, GitHub, and Hugging Face.
>
> *Utilizing your feedback:* We recognize that additional dataset details may be of particular interest to readers already familiar with the challenges in avian bioacoustics and audio classification. However, after careful consideration, we chose to maintain our focus on a beginner-friendly introduction to engage a broader audience across different domains rather than including more dataset specifics in the main text that would require us to remove a lot of valuable specifics from Section 2. That said, we acknowledge your remark and plan to expand both the GitHub and Hugging Face repositories with more detailed dataset information once the review phase concludes, allowing us to provide non-anonymized links.
>
> ### **(2) Results for additional evaluation use cases**
>
> > The results focus mainly on the multi-label classification benchmark, with limited exploration of other use cases. Given the dataset’s availability of precise event timestamps, it would be valuable to include benchmarks for sound event detection. Additionally, providing results for other mentioned use cases would strengthen the paper.
>
> *Explanation and statement:* An important goal of this paper is to introduce the dataset and associated challenges to the ML/DL community, emphasizing its relevance as a (novel) domain-specific resource for audio classification. As you pointed out, our work's broad applicability/flexibility could accommodate numerous additional evaluation use cases - such as sound event detection in soundscapes (L.73). We chose to concentrate on the challenges of multi-label classification since it serves as an important benchmark for both audio classification and the bioacoustics community while remaining accessible/understandable as an initial evaluation use case. Therefore, including further evaluation use cases would exceed the intended scope of this paper, especially given the current page limit constraints.
>
> *Utilizing your feedback:* We agree that adding additional evaluation use cases would certainly broaden the scope of the paper, but it falls outside of our current objectives and would be challenging to implement within the constraints of a short time frame and limited page space. Nevertheless, this is a valuable suggestion, and we plan to expand our Hugging Face leaderboard (Appendix A, L. 940), which currently focuses solely on the multi-label classification use case, to include additional evaluation scenarios such as the event detection you mentioned. Furthermore, we are currently working on another evaluation case that utilizes BirdSet as a source for self-supervised learning to train an AudioMAE-style model for fine-tuning. This will lay the groundwork for future work and a new evaluation case, which we will soon incorporate into the leaderboard, potentially spanning multiple projects derived from BirdSet.

---

> > ### Comment · Reviewer_qFwz · 2024-11-25
> >
> > Thanks to the authors for the detailed and thoughtful responses. I appreciate the effort you have taken to address the points raised, and it addresses my concerns effectively. I believe that with the inclusion of more detailed dataset information and additional evaluation use cases, the proposed dataset has the potential to become an excellent resource for future research.

---

### Official Review · Reviewer_5AN6 · 2024-10-28

**Soundness:** 2
**Presentation:** 3
**Contribution:** 2
**Rating:** 6
**Confidence:** 4

**Summary:**

The paper proposes BirdSet, which is a dataset, or more specifically, a collection of 11 different sub-datasets. BirdSet attempts to unify
avian bioacoustic evaluation on focal and soundscape recordings under one roof, providing a large, unified, accessible test bed for testing audio classification approaches.

**Strengths:**

1. Well written and easy to read.
2. Thoroughly explains the challenges experienced not only in avian bioacoustics, such as covariate shift and mismatch in focal and soundscape recordings, but also in curating, and developing a dataset of such a size and scale.
3. The pain points addressed in the paper are very real: poor availability and accessibility of AudioSet, lack of a unified benchmark suite for evaluating segment and event-based bioacoustics tasks, and mismatch between train and test time for avian bioacoustics, are all pressing challenges.
4. A good variety of baseline models were evaluated.

**Weaknesses:**

To me, the paper, in several places, tries to pose BirdSet as a replacement for AudioSet and that it should be the exemplary benchmark for evaluating audio classification models, with statements such as "Avian bioacoustics exemplifies challenges in audio classification...", and how curated datasets like AudioSet and ESC-50 do not represent real-world complexities. Pain points mentioned w.r.t. AudioSet are all very real, but AudioSet is a much broader dataset than BirdSet. Several people, in industry and academia have found AudioSet useful: a blanket statement saying AudioSet is not useful in real-world scenarios is plain wrong.
Some statements straight up downplay AudioSet: for instance, line 37-38: where AudioSet poses "... concerns regarding transferability to real-world environmental domains". Whereas the paper cited simply shows that AudioSet pretrained models perform "well enough": of course models trained on in-domain bioacoustics data will fare better for bioacoustic evaluation!

Also, as per Table 3, AST and EAT models, which are pretrained on AudioSet, seem to perform quite well in the DT setting (cross-domain transfer setting) versus their LT and MT counterparts. For EAT, DT performance matches MT and outperforms LT scenarios.

BirdSet will be a tremendous contribution to the fields of audio classification and avian bioacoustics, I have no doubt, but I think the current language of the paper comes off as "posturing" a tad bit too much.

**Questions:**

In the context of how the paper is currently phrased, more direct comparisons, for instance, linear evaluation on cross-domain data between models trained on BirdSet and AudioSet on a variety of downstream tasks, spanning bioacoustics and other audio domains would be needed.

---

> ### Author Response · Authors · 2024-11-14
> **Authors' Answer (1): Positioning of BirdSet (1/2)**
>
> Dear reviewer,
> thank you very much for your thorough and insightful review. We are happy that you acknowledge the impactful contribution of BirdSet and share our perspective on the key challenges within the audio classification domain. In the following, we go through your two main remarks, explain our approach and how we have incorporated your valuable feedback.
>
> ### **(1) Positioning of BirdSet compared to AudioSet**
>
> > BirdSet will be a tremendous contribution to the fields of audio classification and avian bioacoustics, I have no doubt, but I think the current language of the paper comes off as "posturing" a tad bit too much.
>
> *Explanation and statement**:* First, we entirely agree that AudioSet was a pivotal step for audio classification and remains a highly relevant and impactful dataset. Our intention with BirdSet was never to replace it but to provide a complementary, domain-specific dataset that is easily accessible and supports a wide range of novel/challenging evaluation use cases. We also agree that datasets like AudioSet/ESC-50 capture real-world challenges, even if they are more curated than BirdSet. We aim to add another layer of benchmarking for audio classification with an ecologically relevant focus.
>
> *Utilizing your feedback**:* We went through the paper to identify and carefully adjust sections that might have overstated BirdSet’s positioning. We aim to maintain the core message and BirdSet’s strong contribution while ensuring its complementary value to the domain. Any change in the paper PDF is highlighted in red (we reupload as soon as all updates are done). We summarize all changes in the following:.
>
> **L.13-16 (Abstract)**:
> >“While AudioSet aims to bridge this gap as a universal-domain dataset, its restricted accessibility and lack of diverse real-world evaluation use cases challenge its role as the primary resource. Therefore, we introduce BirdSet, a large-scale benchmark dataset for audio classification focusing on avian bioacoustics.”
>
> _Changes_:  While AudioSet is a pivotal step to bridge this gap as a universal-domain dataset, its restricted accessibility and limited evaluation use cases challenge its role as the sole resource. Therefore, we introduce BirdSet, a large-scale complementary benchmark data set for audio classification focusing on avian bioacoustics.
>
> **L.34-38 (1. Introduction)**
> >“While AudioSet (Gemmeke et al., 2017) offers substantial training data with over 5,800 recording hours, its restricted accessibility that requires manual data retrieval, lack of diverse evaluation scenarios and test datasets (Wang et al., 2021), and concerns regarding transferability to real-world environmental domains (Ghani et al., 2023) challenge its role as the only training resource.”
>
> _Changes_: Although AudioSet (Gemmeke et al., 2017) offers substantial training data with 5,800 recording hours, its restricted accessibility that requires manual data retrieval, lack of diverse evaluation scenarios (Wang et al., 2021), and studies emphasizing the need for domain-specific representation learning (Ghani et al., 2023), challenge its role as the sole training resource.
>
> _Further Explanations_: In this part, we have already emphasized that AudioSet should not be replaced. Instead, we highlighted the necessity of additional resources beyond AudioSet in the future and the importance of having diverse and versatile datasets. We hope our clarifications have further reinforced this perspective. However, we would like to maintain the wording regarding the limited use cases and test dataset availability, as this highlights a unique aspect of BirdSet — its seven different evaluation datasets, each with distinct traits and features that introduce unique domain transfer challenges not inherently present in AudioSet. Additionally, our test datasets are accompanied by strong labels, minimizing potential labeling errors often encountered when evaluating with AudioSet.
>
> Regarding in-domain transferability: Paper [1] explicitly mentions that conventional supervised learning in avian bioacoustics achieves better transferability than more general self-supervised learning representations from AudioSet. You are correct that this reinforces the expectation that in-domain transfer outperforms out-of-domain transfer, but there is a discrepancy between SL and SSL representation learning that is very interesting to research in the future: We are currently developing an AudioMAE-style SSL model with BirdSet. Preliminary results demonstrate improved transferability (in contrast to [1]) when pre-training on AudioSet and fine-tuning on BirdSet and even better fine-tuning performance with BirdSet's in-domain SSL training.
>
> [1] Burooj Ghani, Tom Denton, Stefan Kahl, and Holger Klinck. Feature Embeddings from Large-Scale Acoustic Bird Classifiers Enable Few-Shot Transfer Learning. CoRR, 2023. URL https: [//doi.org/10.48550/arXiv.2307.06292](https://doi.org/10.48550/arXiv.2307.06292).

---

> ### Author Response · Authors · 2024-11-14
> **Authors' Answer (2): Positioning of BirdSet (2/2)**
>
> **L.76 (1. Introduction - Contributions)**
> > "[…] BirdSet represents a comprehensive resource in audio classification.”
>
> _Changes_: […] BirdSet represents a comprehensive and additional resource in audio classification.
>
> **L.91, 92-93, 101-102 (2. Current Challenges and Related Work)**
> >> “Avian bioacoustics exemplifies challenges in audio classification, including managing diverse and noisy acoustic environments or dealing with class imbalance.”
> >
> >> “[…] it serves as our primary case study for illustrating real-world evaluation use cases […].”
> >
> >> “As a domain-specific audio classification task, avian bioacoustics exemplifies and adds to these complexities, making it suitable for exploring real-world audio challenges.”
>
> _Explanations_: We did not change anything here but since you explicitly highlighted L.91 and the other parts go into the same directions, we want to clarify that we intended to convey that avian bioacoustics comes with unique traits that present interesting and challenging opportunities for the broader audio classification domain. This was not meant to undermine the value of AudioSet but to illustrate why avian bioacoustics is suitable as our primary case study and comes with relevant real-world evaluation use cases.
>
> **L.142-146 (2.1 Challenge 1: Datasets)**
> >“[…] However, these curated datasets do not fully represent real-world complexities (e.g., class imbalance) and are considerably smaller compared to vision datasets (e.g., ImageNet (Deng et al., 2009)).”
>
> _Changes_: However, these datasets cannot fully represent real-world complexities (e.g., class imbalance) due to limited size and quantity compared to vision datasets (e.g., ImageNet (Deng et al., 2009)).
>
> **L. 158-164 (2.1 Challenge 1: Datasets)**
>
> _Explanations_: We believe this aspect of the related work adequately highlights some critical differences from AudioSet without implying that it is intended to replace it altogether.
>
> **L. 191-193 (2.2 Challenge 2: Model Training)**
> >“This complexity is often reduced to a more straightforward multi-class task in datasets such as ESC-50 (Piczak, 2015), limiting a model’s real-world applicability where overlapping sounds and event ambiguity are common.”
>
> _Changes_: […], potentially limiting a model's ability to handle overlapping sounds and event ambiguity in practice.
>
> **L. 224-226 (2.2 Challenge 2: Model Training)**
> > “In contrast, models pre-trained on general domain audio exhibit poor transferability to the domain (Ghani et al., 2023; Nolan et al., 2023)”
>
> _Changes_: In contrast, pre-training on general domain audio through SSL exhibits limited transferability, compared to in-domain transfer of supervised models (Ghani et al., 2023).

---

> ### Author Response · Authors · 2024-11-14
> **Authors' Answer (3): Cross-Domain Evaluation (1/1)**
>
> ### **(2) Cross-Domain Evaluation**
>
> >In the context of how the paper is currently phrased, more direct comparisons, for instance, linear evaluation on cross-domain data between models trained on BirdSet and AudioSet on various downstream tasks spanning bioacoustics and other audio domains, would be needed.
>
> *Explanation and statement**:* With the abovementioned changes from (1) to tone down the implications of replacing existing datasets, we believe this improves the paper. While we understand and appreciate this remark, we feel that these additions would exceed the paper’s scope in its current form after carefully updating the wording. Additionally, we acknowledge that this topic is indeed very interesting for future exploration, as stated in Limitation L.538, where we conclude that a more thorough investigation into the influence of pre-training remains an important limitation and an intriguing direction for future work. Your spotting further highlights this regarding the performance of AST and EAT after fine-tuning, which are pre-trained on AudioSet.
>
> *Utilizing your feedback*: With the changes to the wording and dataset positioning, your remark presents an exciting direction for future work branching out from BirdSet. We believe this work lays the groundwork for these investigations, such as the impact of cross-domain pre-training and fine-tuning.

---

> > ### Comment · Reviewer_5AN6 · 2024-11-15
> > **Response to Author's**
> >
> > Thanks to the authors for the rebuttal. I appreciate the proposed changes to the language: I believe it only solidifies the author's case and strengthens BirdSet's positioning in the eyes of the community.
> >
> > I understand the authors' comments regarding linear evaluation on cross-domain data going beyond the scope of the paper: it's difficult to do experiments while being thorough in a limited amount of time. However, I continue to feel like they would have been instrumental in understanding the nuances of cross-domain transfer and the impact of BirdSet.
> >
> > In light of the above, I have increased my recommendation to 6.

---

### Official Review · Reviewer_uitS · 2024-11-03

**Soundness:** 4
**Presentation:** 4
**Contribution:** 4
**Rating:** 8
**Confidence:** 5

**Summary:**

This paper presents BirdSet, a new large-scale benchmark dataset specifically for multi-label audio classification within avian bioacoustics. It significantly extends the scope of existing audio datasets by including approximately 10,000 classes, covering over 6,800 hours of training data, and incorporating 400 hours across eight distinct test datasets. BirdSet addresses several real-world machine learning challenges, such as covariate shift, label noise, and task shift, providing a unique resource for evaluating model robustness in audio classification under diverse conditions. The benchmark includes evaluations of six prominent deep learning models across three different training approaches: training on the full BirdSet, training on a subset containing only classes relevant to the downstream tasks, and training on a small subset for each downstream dataset individually.

Additionally, the authors facilitate accessibility by hosting BirdSet on Hugging Face, where they provide Python code to load the data. They also offer scripts for reproducing the experiments in the paper​.

**Strengths:**

1. Originality

BirdSet represents a novel contribution to multi-label audio classification. With close to 10,000 classes BirdSet provides a benchmark to develop scalable methods capable of handling extreme class diversity with large imbalance. BirdSet also addresses critical machine learning challenges such as covariate shift, where the testing distribution diverges from the training distribution reflecting real-world environmental shifts in field data.


2. Clarity

The paper is clear and well-organized, with a logical presentation of BirdSet’s structure, design choices, and challenges. The descriptions of covariate and domain shifts, as well as the evaluation protocols and training setups, are concise and well-articulated.


3. Significance

BirdSet can enable the development of new self-supervised learning, active learning, and few-shot learning approaches that are resilient to covariate and domain shifts—capabilities that are increasingly relevant for practical deployment of models in real-world applications. Moreover, its emphasis on multi-label classification reflects real-world scenarios in bioacoustics and similar domains, driving advancements that can translate to other fields beyond audio.


In summary, BirdSet is highly original, well-curated, and impactful dataset that serves as both a resource and a benchmark. It encourages tackling challenges related to robustness to distribution shifts, and multi-label classification, with methods that can be developed that extend across deep learning domains more broadly.

**Weaknesses:**

I didn’t find any weaknesses in this paper.

**Questions:**

I see significant potential in BirdSet, and I wonder if it would be feasible to define few-shot tasks within the downstream tasks to address few-shot multi-label classification?

---

> ### Author Response · Authors · 2024-11-15
> **Authors' Answer: Few-shot Scenarios**
>
> Dear reviewer, thank you for your detailed review and very positive feedback! We are very happy that you recognize the significance of our work. As no specific weaknesses were highlighted, we will focus on addressing your interesting question below:
>
> > I wonder if it would be feasible to define few-shot tasks within the downstream tasks to address few-shot multi-label classification?
>
> This is an excellent question and highlights an interesting direction for future evaluation use cases, especially in the context of model adaption (Challenge 3 and contribution (2) in the paper), which is fundamental for advancing practical applications in audio (avian) classification. As noted in Line 320, BirdSet’s flexible and versatile design opens up numerous potential tasks and scenarios. This is particularly relevant when fine-tuning a model that was e.g. pre-trained through self-supervised learning (SSL) (which is still largely unexplored within the field of avian bioacoustics). Here are two examples with which we could utilize the few-shot multi-label classification with BirdSet:
>
> - One possible approach is to train a model using SSL (or SL) on our large dataset XCL and then create few-shot scenarios within the dedicated subsets for fine-tuning (e.g., in the direction of [1] or [2]), where a small number ($k$) of examples are selected from the respective classes in the training subsets. This could create multiple real-world few-shot scenarios when evaluated on our test datasets. Currently, we are developing an AudioMAE-style SSL model pre-trained XCL and fine-tuning it on relevant subsets. As the preliminary results are quite promising and show potential for valuable representations, we could seamlessly extend it into few-shot scenarios, making your suggestion particularly timely and valuable!
> - Another approach could also be to directly utilizing the test soundscape recordings through a form of cross-validation to fine-tune a pre-trained model in a few-shot setting, specifically focusing on a subset of bird species (or only a single species) within these soundscapes (which could also be relevant for practitioners).
>
> [1] Burooj Ghani, Tom Denton, Stefan Kahl, and Holger Klinck. Feature Embeddings from Large Scale Acoustic Bird Classifiers Enable Few-Shot Transfer Learning. CoRR, 2023. URL https:[//doi.org/10.48550/arXiv.2307.06292](https://doi.org/10.48550/arXiv.2307.06292)
>
> [2] Jenny Hamer, Eleni Triantafillou, Bart Van Merriënboer, Stefan Kahl, Holger Klinck, Tom Denton, and Vincent Dumoulin. BIRB: A Generalization Benchmark for Information Retrieval in Bioacoustics. CoRR, 2023. URL https://doi.org/10.48550/arXiv.2312.07439

---

> > ### Comment · Reviewer_uitS · 2024-11-15
> >
> > Thank you to the authors for taking the time to address my question and for providing such a thoughtful explanation of possible directions for few-shot learning in a multi-label setting. These approaches present exciting possibilities and open up promising avenues for both practical applications and future research.

---

### Official Review · Reviewer_wpw3 · 2024-11-07

**Soundness:** 4
**Presentation:** 4
**Contribution:** 3
**Rating:** 8
**Confidence:** 4

**Summary:**

This is an excellent,  comprehensive and carefully curated dataset of bird sounds. The paper also provides an excellent review of the existing literature, and provides baseline code showing use of the data. The benchmark testing tasks are a bit simple, but demonstrate the breadth of the dataset.

**Strengths:**

Careful curation of the dataset - types of collections, varying types of birdcalls from same species; includes both soundscapes, and individual recordings; comparisons with other datasets.
Inclusion of benchmark code for future researchers to use as a baseline.
Permissive licensing
A large scale project

**Weaknesses:**

too many uncommon acronyms which require a reader to keep going back and forth - such shortening of the length was unnecessary. The acronyms are used in the figures as well. Figures and captions are supposed to stand on their own.

**Questions:**

Please rewrite with clarity and minimizing acronyms.

---

> ### Author Response · Authors · 2024-11-16
> **Authors' Answer: Acronym Usage**
>
> Dear reviewer, thank you for your insightful review and very positive feedback. We appreciate your acknowledgment of BirdSet's scope and potential for future research. Below, we directly address your main remark and outline how we have leveraged it to enhance the paper:
>
> ### **Acronym Usage**
>
> > too many uncommon acronyms which require a reader to keep going back and forth - such shortening of the length was unnecessary. The acronyms are used in the figures as well. Figures and captions are supposed to stand on their own.
>
> *Explanation and statement:* Our intention with the acronyms was to enhance the reading experience, but we understand that this may lead to difficulties if readers need to frequently jump back and forth within the paper. Regarding the recommendation that every figure/table should stand on its own, we chose to include explanations within the text to allow for larger tables/figures and enhance their visibility. However, we recognize that this approach may necessitate readers to refer back to the text for context, potentially making the reading experience more challenging.
>
> *Utilizing Your Feedback:* We went through the paper to identify challenging abbreviations and to ensure figure/table captions can be understood independently from the text.  While you did not specify the acronyms, we focused on the experimental section, where we introduce our experimental settings (DT, MT, LT) and corresponding figures/tables. Removing these abbreviations completely would require removing substantial content due to space constraints. Although this could enhance reading flow, it would compromise the paper's overall quality by requiring text reduction or figure downsizing. Thus, we opted for a middle-ground solution: introducing the scenarios in bold and incorporating them directly into figures and tables. We believe that this way we can maintain the content and improve readability. Below, we list the specific changes in the following. All modifications in the paper are marked in blue (the paper is updated here as soon as we worked in all changes).
>
> Table 3 and Figure 6 ( L.493 in 4.2 Empirical Results) have been updated to be self-explanatory, eliminating the need to reference the text for the abbreviations DT, MT, and LT. Additionally, we have highlighted the training scenarios (L.445, L.447 in 4.1 Experimental Setup) in bold to enable quick reference if necessary and improve readability.
>
> **L.445, 447 (4.1 Experimental Setup)**
>
> *Changes*: We now introduce the training scenarios in **bold**.
>
> **Figure 6 (4.2 Empirical Results)**
>
> >Mean AUROC in LT with selected models across test datasets.
>
> *Changes*: Mean AUROC in large training (LT) with selected models across test datasets.
>
> **Table 3 (4.2 Empirical Results)**
>
> > Mean results across datasets and models for LT. Best and second best results are highlighted. Score implies the test average for all training scenarios (DT, MT, LT).
>
> *Changes:* Mean results across datasets and models for large training (LT) on XCL. Best and second best results are highlighted. Score reflects test average for all training scenarios, also including dedicated fine-tuning (DT) on subsets, medium training on XCM.
>
> **Table 2 in Section 3**
> > Overview of datasets in BirdSet.
>
> *Changes:* Overview of dataset statistics in BirdSet. J denotes the evenness index.
>
> Regarding our dataset abbreviations: writing them out in full would  significantly increase space requirements. Since they are thoroughly explained in the appendix, and their detailed names are not critical for understanding the main text, we’d like to retain the use of abbreviations in this context. Thus, we would stick with abbreviations in Figure 1 and Table 1 and dataset abbreviations in the text. The same goes for abbreviations for models: if we would write them out each time, we would exceed the page limit quite fast (e.g., AST).

---

### Author Response · Authors · 2024-11-25
**Authors' General Comment and Summary**

Dear reviewers,

Thank you again for your detailed feedback, which has improved the quality of our paper. We also appreciate your recognition of the paper's strengths, including its clarity and high quality [uitS, wpw3, 5AN6, qFwz]. Specific highlights include:
- *Accessibility and curation*: The dataset's availability via Hugging Face and its careful curation, along with benchmark code [uitS, wpw3].
- *Thorough explanation of challenges*: Structured discussions on audio classification challenges, particularly AudioSet's accessibility and bioacoustics' training-testing data mismatch [uitS, 5AN6, qFwz], as well as an intensive review of related work [wpw3].
- *Impact of BirdSet and its contributions*:
    - “BirdSet will be a tremendous contribution to the fields of audio classification and avian bioacoustics” [5AN6]“
    - “The introduction of *BirdSet* fills a notable gap in audio classification by providing a large-scale, domain-specific dataset for avian bioacoustics” [qFwz]
    - “BirdSet is highly original, well-curated, and impactful dataset that serves as both a resource and a benchmark.” [uitS]
    - “This is an excellent, comprehensive and carefully curated dataset of bird sounds.” [wpw3]

**Opportunities for Improvement**

All revisions in the main paper are highlighted in blue [wpw3] and red [5AN6]. While we have addressed most of the suggestions, certain out-of-scope or page-limit-related requests required compromises. The rationale behind our decisions is explained in detail in the specific rebuttals.

- *Acronym usage and figure/table captions*: Reviewer wpw3 noted uncommon acronyms, particularly in figures and tables, thar required cross-reading. We revised the paper to introduce abbreviations (DT, MT, LT) more clearly in 4.1 Experimental Setup and restructured 4.2 Empirical Results figures/tables' captions to be standalone. These changes, marked in blue, ensure the results are easier to follow without frequent cross-referencing.

- *Positioning of BirdSet*: Reviewer 5AN6 expressed concerns about BirdSet's positioning relative to AudioSet. To clarify, we revised the text (highlighted in red) to present BirdSet as a complementary rather than a replacement dataset. This more clearly conveys the overall message and addresses concerns about overstating its scope, as noted positively by reviewer 5AN6.

- *Expanded dataset description and evaluation use cases*: Reviewer qFwz suggested expanding the dataset description and adding evaluation use cases. While these suggestions were valuable, page and time constraints limited our ability to include them in detail. Instead, we committed to providing detailed statistics on GitHub and Hugging Face after the double-blind review phase and introducing an additional use case on the leaderboard. These plans were positively noted.

**Summary**

Although our detailed rebuttals address each review individually, this general comment summarizes our main changes, emphasizing how they enhance the paper while preserving its core message: introducing BirdSet as a novel, multi-purpose dataset for audio classification with various evaluation use cases as an domain-specific alternative to AudioSet. We hope these revisions demonstrate our commitment to addressing your feedback thoughtfully. If you have any further questions, we would be happy to discuss them.

---

### Meta-Review · Area_Chair_3HpH · 2024-12-21

**Metareview:**

The paper proposes a  benchmark dataset  for multi-label audio classification in bird acoustics. This dataset  improve upon existing audio datasets like audioset,  by proposing approximately thousands of classes and different test datasets.
The new dataset aims at addressing several challenges in machine learning, including domain adaptation shift, label noise, and task shift, making it an excellent resource for assessing model robustness in diverse audio classification scenarios. The benchmark evaluates deep learning models using three different training strategies.  All reviewers are happy to accept the paper.

**Additional Comments On Reviewer Discussion:**

all reviewers were happy about the rebuttals.

---

### Decision · Program_Chairs · 2025-01-22

Accept (Spotlight)